# *T-REX17* is a transiently expressed non-coding RNA essential for human endoderm formation

Alexandro Landshammer[1,2†], Adriano Bolondi[1,2†], Helene Kretzmer[1], Christian Much[3], René Buschow[4], Alina Rose[5], Hua-Jun Wu[6,7], Sebastian D Mackowiak[1], Bjoern Braendl[1], Pay Giesselmann[1], Rosaria Tornisiello[1], Krishna Mohan Parsi[8], Jack Huey[8], Thorsten Mielke[4], David Meierhofer[9], René Maehr[7,10], Denes Hnisz[1], Franziska Michor[11,12,13,14], John L Rinn[3], Alexander Meissner[1,2,11,12]*

[1]Department of Genome Regulation, Max Planck Institute for Molecular Genetics, Berlin, Germany; [2]Institute of Chemistry and Biochemistry, Freie Universität Berlin, Berlin, Germany; [3]Department of Biochemistry, University of Colorado Boulder and BioFrontiers Institute, Boulder, United States; [4]Max Planck Institute for Molecular Genetics, Microscopy Core Facility, Berlin, Germany; [5]Helmholtz Institute for Metabolic, Obesity and Vascular Research, Leipzig, Germany; [6]Department of Data Science, Dana-Farber Cancer Institute, Department of Biostatistics, Harvard T. H. Chan School of Public Health, Boston, United States; [7]Center for Precision Medicine Multi-Omics Research, School of Basic Medical Sciences, Peking University Health Science Center and Peking University Cancer Hospital and Institute, Beijing, China; [8]Program in Molecular Medicine, University of Massachusetts Medical School, Worcester, United States; [9]Max Planck Institute for Molecular Genetics, Mass Spectrometry Core Facility, Berlin, Germany; [10]Diabetes Center of Excellence, University of Massachusetts Medical School, Worcester, United States; [11]Department of Stem Cell and Regenerative Biology, Harvard University, Cambridge, United States; [12]Broad Institute of MIT and Harvard, Cambridge, United States; [13]Department of Data Science, Dana-Farber Cancer Institute, and Department of Biostatistics, Harvard T. H. Chan School of Public Health, Boston, United States; [14]The Ludwig Center at Harvard, Boston, MA 02215, USA, and Center for Cancer Evolution, Dana-Farber Cancer Institute, Boston, United States

*For correspondence: meissner@molgen.mpg.de

†These authors contributed equally to this work

**Abstract** Long non-coding RNAs (lncRNAs) have emerged as fundamental regulators in various biological processes, including embryonic development and cellular differentiation. Despite much progress over the past decade, the genome-wide annotation of lncRNAs remains incomplete and many known non-coding loci are still poorly characterized. Here, we report the discovery of a previously unannotated lncRNA that is transcribed 230 kb upstream of the *SOX17* gene and located within the same topologically associating domain. We termed it *T-REX17* (Transcript Regulating Endoderm and activated by soX17) and show that it is induced following SOX17 activation but its expression is more tightly restricted to early definitive endoderm. Loss of *T-REX17* affects crucial functions independent of SOX17 and leads to an aberrant endodermal transcriptome, signaling pathway deregulation and epithelial to mesenchymal transition defects. Consequently, cells lacking the lncRNA cannot further differentiate into more mature endodermal cell types. Taken together, our study identified and characterized *T-REX17* as a transiently expressed and essential non-coding regulator in early human endoderm differentiation.

## Editor's evaluation

Supported by a large set of complementary experiments, the authors convincingly show that the lncRNA *T-REX17* is required for human definitive endoderm differentiation. *T-REX17* function is not related to the adjacent *SOX17* gene that lies in the same topological domain (TAD), implying a trans-acting role. The study is important because it sheds light on the stage-specific role of lncRNAs in cell lineage induction.

## Introduction

To date, nearly 28,000 long non-coding RNAs (lncRNAs) have been reported in the human genome, but less than 1% (~150) has been functionally characterized (*Ransohoff et al., 2018*; *Hon et al., 2017*; *Quek et al., 2015*; *Jiang et al., 2016*). Several of those have been shown to influence cellular physiology in developmental, adult and disease contexts (*Sarropoulos et al., 2019*; *James, 2015*; *Prensner et al., 2011*; *Castellanos-Rubio et al., 2016*; *Perry and Ulitsky, 2016*; *Lorenzi et al., 2021*). Depending on their genomic location, lncRNAs can be classified into genic lncRNAs (overlapping with a protein-coding gene) or intergenic lncRNAs (lincRNAs; no overlap with a protein-coding gene) (*Ransohoff et al., 2018*). Together with transcription factors and epigenetic regulators (*Hung et al., 2011*; *Jeon and Lee, 2011*; *Boque-Sastre et al., 2015*), lncRNAs participate in complex gene-regulatory networks by fine-tuning gene expression in a precise and controlled manner (*Grote and Herrmann, 2015*). In particular, lncRNAs have been shown to modulate gene expression at multiple levels, including chromatin structure and folding (*Gupta et al., 2010*), activating neighboring (*Engreitz et al., 2013*) and distal (*Hacisuleyman et al., 2014*) genes, affecting RNA splicing (*Pisignano and Ladomery, 2021*), or influencing nuclear compartmentalization (*Caudron-Herger and Rippe, 2012*; *Rinn and Guttman, 2014*; *Quinodoz and Guttman, 2014*).

More specifically, long non-coding RNAs have also been shown to fine-tune the activation and function of developmental regulators, including transcription factors responsible for maintenance of pluripotency (*Sheik Mohamed et al., 2010*; *Ulitsky et al., 2011*; *Ng et al., 2012*), mesoderm specification (*Frank et al., 2019*) and neuronal differentiation (*Xi et al., 2022*). Recent studies have also attributed critical roles for lncRNAs in the early stages of human development, in particular during definitive endoderm specification through *cis*-regulatory activity on nearby genes (*Jiang et al., 2015*; *Yang et al., 2020*). For instance, *LNC00261* facilitates the activation of the proximal *FOXA2* gene via association with SMAD2/3 (*Jiang et al., 2015*). A mechanistically similar *cis*-regulation of *GATA6* has been attributed to lncRNA *GATA6-AS1* (*Yang et al., 2020*), while the lncRNA *DIGIT* has been reported to control *GSC* in *trans*-, via the formation of BRD3-dependent phase-separated condensates (*Daneshvar et al., 2016*; *Daneshvar et al., 2020*). The majority of lncRNAs exhibit highly tissue-specific expression, often more restricted than observed for protein-coding genes (*Cabili et al., 2011*). Signaling molecules, including TGF-β, WNT and the JUN/JNK/AP-pathway represent critical cascades necessary for endoderm formation, inducing the expression of endodermal factors such as SOX17, GATA6 and C-X-C chemokine receptor 4 (CXCR4) (*Li et al., 2019*; *Chia et al., 2019*; *Fisher et al., 2017*). SOX17 is a member of the SOX-F group of transcription factors and its expression is necessary for the specification of definitive endoderm in vitro (*Séguin et al., 2008*) and in vivo (*Kanai-Azuma et al., 2002*). Despite being an essential and well-studied gene, much remains to be understood about the regulatory elements and nuclear organization of the larger *SOX17* domain and how it functions in early endoderm development.

## Results

### Discovery of an unannotated non-coding transcript within the *SOX17* topological domain

So far, *SOX17* is the only annotated gene located within the 336 kb *SOX17* loop-domain insulated by strong CTCF-boundaries (*Figure 1A*, top). However, upon closer inspection of multiple epigenetic modifications in pluripotent stem cells (hESCs and hiPSCs) and early definitive endoderm we observed a potential unannotated gene locus. In particular, the combination of histone H3 lysine 4 trimethylation

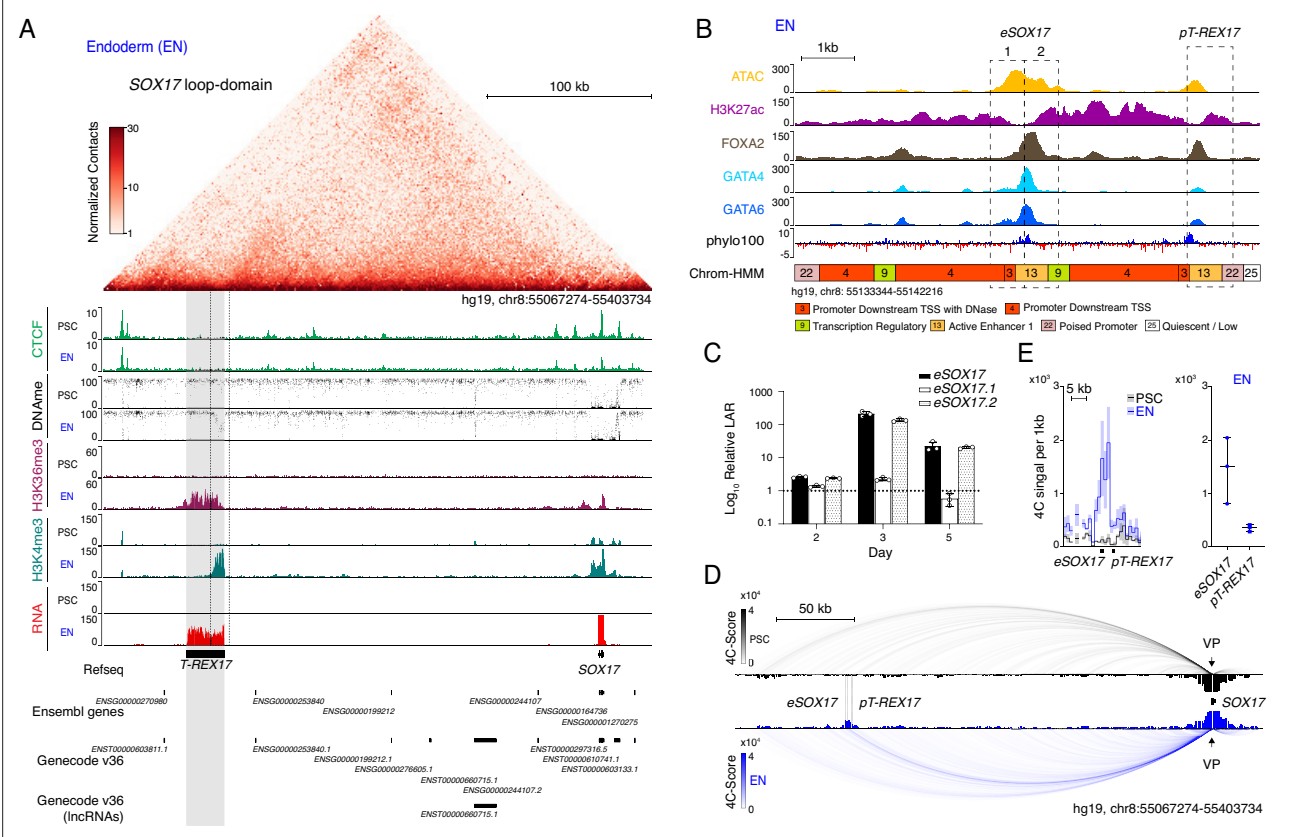

**Figure 1.** Identification of *T-REX17* at the human *SOX17* locus. (**A**) Normalized capture Hi-C (cHi-C) contact map of the human *SOX17* locus in endoderm cells (top panel) and chromatin immunoprecipitation sequencing (ChIP-seq) tracks of CTCF, H3K36me3 and H3K4me3 as well as whole genome bisulfite sequencing (WGBS) (*Supplementary file 1*) and RNA-seq profiles in PSCs and EN (bottom panel). *T-REX17* locus (hg19, chr8:55117776–55140806) is highlighted in grey. (**B**) Zoomed in view of the *SOX17* distal regulatory element in EN cells comprising Assay for Transposase-Accessible Chromatin with high-throughput sequencing (ATAC-seq) profile and H3K27ac, FOXA2, GATA4 and GATA6 ChIP-seq (*Supplementary file 1*) profiles. Chrom-HMM (*Ernst and Kellis, 2015*; *The ENCODE Project Consortium, 2012*) 25-state profile is shown below the phylo100 (*Murphy et al., 2001*; *Pollard et al., 2010*) UCSC conservation track. Dashed lines indicate the two distinct regulatory elements, characterized by enriched transcription factors occupancy (*eSOX17* and *pT-REX17*). (**C**) Firefly luciferase assay from either *eSOX17.1* (hg19, chr8:55136923–55137557), *eSOX17.2* (hg19, chr8:55137558–55138192) or both together at days 2, 3, or 5 of EN differentiation. Values are calculated as luciferase activity ratio (LAR) between firefly and renilla signal, normalized on empty vector background and day 0 baseline signal. Bars indicate mean values, error bars show standard deviation (SD) across three independent experiments. Individual data points are displayed. Raw measurements are reported in *Supplementary file 1*. (**D**) 4Cseq of PSC (black) and EN (blue) at the *SOX17*-locus. Normalized interaction-scores displayed as arcs and histogram-profiles utilizing the *SOX17* promoter as viewpoint (VP). (**E**) 4Cseq interactions as a zoomed in view at the *SOX17* regulatory element and corresponding quantification. In the zoomed in tracks, the line represents the median and the shaded areas depict 95% CI; in the quantification, the central line represents the median and error bars show SD across three independent experiments.

The online version of this article includes the following source data and figure supplement(s) for figure 1:

**Figure supplement 1.** Functional characterization of the *SOX17* distal regulatory elements.

**Figure supplement 1—source data 1.** Source data for the genotyping gel in *Figure 1—figure supplement 1E*.

(H3K4me3) and histone H3 lysine 36 trimethylation (H3K36me3) in ESC-derived endoderm suggested the presence of an RNA Polymerase-II-driven transcript (*Guttman et al., 2009*; *Bilodeau et al., 2009*). Further supporting this, matched RNA sequencing data showed a 22 kb long transcribed region approximately 230 kb upstream of *SOX17* (*Figure 1A*, bottom). These results combined with a strong UCSC PhyloCSF sequence conservation points to an intergenic lncRNA (lincRNA) that we subsequently termed *T-REX17* (Transcript Regulating Endoderm and activated by soX17) (*Figure 1A and B*). Although the sequence conservation to the mouse is only modest (*Figure 1—figure supplement 1*), we detect the presence of a distal *SOX17* transcript in a number of vertebrates based on stage- and tissue-matched embryonic data (*Figure 1—figure supplement 1A*).

We began to explore the locus in more detail by investigating the overlapping distal regulatory element that appears to be a putative *SOX17* enhancer (*Tsankov et al., 2015*). We found two distinct sites with notable transcription factor (TF) occupancy within a region of open chromatin specifically in definitive endoderm (*Figure 1B*). Although both sites show enriched UCSC PhyloCSF sequence conservation, they are also characterized by a distinguishable promoter and enhancer signature (ChromHMM state 22 and ChromHMM state 13, respectively) (*Figure 1B*; *Ernst and Kellis, 2012*; *Ernst and Kellis, 2017*).

We next assessed the activity of the putative promoter region of *T-REX17* (*pT-REX17*) in a luciferase assay and found it to be endoderm-specific (*Figure 1—figure supplement 1B*). We similarly tested the activity of the putative enhancer, which was further separated into two parts based on its TF occupancy profile (*eSOX17.1* and *eSOX17.2*) (*Figure 1B and C*). The entire region but also *eSOX17.2* alone showed strong enhancer activity during endoderm differentiation (*Figure 1C*, *Figure 1—figure supplement 1C*).

We then further evaluated *eSOX17.2* function using Cas9-induced homozygous deletions and assessed the effect of the mutation during directed endoderm differentiation (*Figure 1—figure supplement 1D and E*). Interestingly, we observed a delayed activation of SOX17 and overall reduced expression of the transmembrane C-X-C chemokine receptor 4 (CXCR4) (*Figure 1—figure supplement 1F and G*). To investigate the physical interactions at the locus, we performed Circularized Chromosome Conformation Capture sequencing (4C-seq) on pluripotent cells and early endoderm and found an enriched interaction between the *SOX17* promoter and its distal enhancer (*eSOX17*) (*Figure 1D and E*; *Figure 1—figure supplement 1H*). Therefore, we can conclude that the topologically isolated domain of *SOX17* encompasses a distal, transcribed region driven by a promoter in close proximity but otherwise independent from a functional enhancer that interacts with the *SOX17* gene.

## *T-REX17* is a definitive endoderm-specific lncRNA

We next investigated the expression of the non-coding transcript during endoderm differentiation with time-resolved qRT-PCR and found that *T-REX17* expression follows *SOX17* kinetics but with an approximate 24 hour delay (*Figure 2A*). To explore possible regulatory links between *SOX17* and *T-REX17*, we compared their expression across a wide range of cell and tissue types (n=44) (*Figure 2B*). *T-REX17* appears tightly restricted to early human definitive endoderm and, importantly, uncoupled from the much broader expression of *SOX17* in many other endoderm-derived tissues (*Mathias et al., 2015*, *Thul et al., 2017*; *Figure 2B*; *Figure 2—figure supplement 1A–C*). Moreover, we utilized RNA-seq data from the three pluripotent stem cell-derived germ layers to show that *T-REX17* is not expressed during mesoderm and ectoderm formation (*Figure 2—figure supplement 1D*). scRNAseq data in the early human gastrulating embryo (*Tyser et al., 2021*) confirms *T-REX17's* tissue specificity in vivo (*Figure 2—figure supplement 1E*).

We also investigated *T-REX17* localization by single-molecule RNA fluorescence in situ hybridization (smRNA-FISH) and found it highly enriched at foci within the nuclear compartment, a characteristic feature of non-coding transcripts (median of 40 foci/cell, *Figure 2C and D*). Nuclear localization and association with chromatin were further confirmed by cell-fractionation experiments (*Figure 2—figure supplement 1F*). Next, we wanted to more closely inspect the coding potential of *T-REX17* and used PhyloCSF to show that 37 of 40 predicted open reading frames (ORFs) would likely result in no functional protein (*Figure 2E*). This is comparable to other short ORFs (sORFs) in the human lncRNA catalog (*Figure 2E*; *Lin et al., 2011*). Notably, even the coding potential of the remaining three sORFs is about two orders of magnitude lower than for the *SOX17* coding sequence (*Figure 2E*).

To explore the structure and splicing variants of *T-REX17,* we used long-read Nanopore sequencing of definitive endoderm cDNA. The two most prevalent isoforms account for 23.3% of the split-reads, while 76.7% appear inconsistently spliced, a feature which is frequently observed in lncRNAs (*Mukherjee et al., 2017*; *Lagarde et al., 2017*; *Schlackow et al., 2017*; *Struhl, 2007*; *Beck et al., 2016*) (termed 'sloppy' splicing, *Figure 2F*; *Figure 2—figure supplement 1G*). Additionally, we used 5' and 3' rapid amplification of cDNA end (RACE) to determine the exact transcriptional start and end sites as well as the corresponding polyadenylation signal (*Figure 2F*; *Figure 2—figure supplement 1H*).

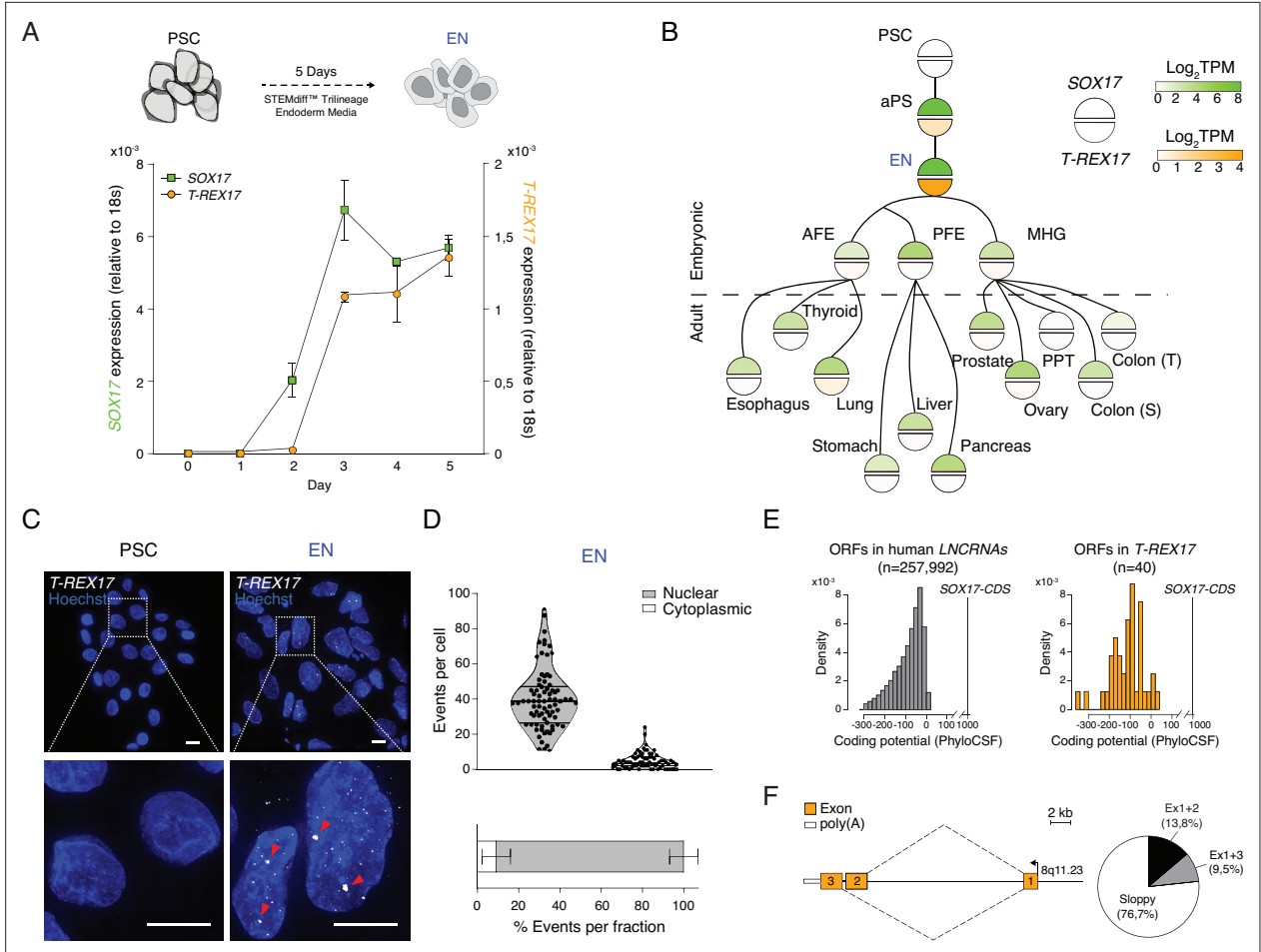

**Figure 2.** *T-REX17* cellular and molecular characterization. (**A**) Time resolved qRT-PCR profiling *SOX17* (green) and *T-REX17* (orange) transcript levels during endoderm differentiation (normalized to the housekeeping gene 18s). Symbols indicate the mean and error bars indicate SD across three independent experiments. (**B**) Lineage tree heatmap showing *SOX17* (green) and *T-REX17* (orange) expression across EN derived embryonic and adult tissues as measured by RNA-seq, extracted from a curated data set of the Roadmap Epigenome Project (***Roadmap Epigenomics Consortium et al., 2015***; ***Supplementary file 1***). TPM, transcripts per million. aPS, anterior primitive streak; AFE, anterior foregut endoderm; PFE, posterior foregut endoderm; MHG; mid-hindgut; PPT, Peyer's patch tissue; S, sigmoid; T, transverse. (**C**) smRNA-FISH of *T-REX17* in PSCs (left) and EN cells (right) counterstained with Hoechst. Red arrowheads indicate two brighter and bigger foci present in each cell, potentially representing sites of nascent transcription. Scale bars, 10 μm. (**D**) Frequencies of *T-REX17* smRNA-FISH foci in the nuclear (grey) or the cytoplasmic (white) compartments. n=79, number of analyzed cells. Lines of the violin plot indicate interquartile range around the median value. In the stacked barplot, error bars indicate SD around the mean value. (**E**) Barplots showing coding potential scores of randomly sampled *LNCRNA* ORFs (n=257,992) (grey) versus *T-REX17* ORFs (n=40) (orange). Scores are shown on the x-axis while ORF-density is plotted on the y-axis. Both conditions area is equal and compared to *SOX17* ORFs as coding gene control. n, number of analyzed ORFs. (**F**) Schematic of *T-REX17* isoform structure derived from MinION-seq reads of endoderm cDNA. Exons are shown in orange while the poly(A) is shown in white. The arrow indicates the transcriptional start site (TSS). Pie chart shows isoform reads (Ex1+2 black n=16, Ex1+3 grey n=11) and 'sloppy spliced' (white n=89) transcript distribution as measured by MinIONseq (***Supplementary file 1***).

The online version of this article includes the following source data and figure supplement(s) for figure 2:

**Figure supplement 1.** *T-REX17* tissue distribution and structural characterization.

**Figure supplement 1—source data 1.** Source data for the cell fractionation assay in ***Figure 2—figure supplement 1F***.

**Figure supplement 1—source data 2.** Sanger sequencing files for the 5′ and 3′ RACE experiment in ***Figure 2—figure supplement 1H***.

Taken together, our results show that *T-REX17* is specifically and transiently expressed in early definitive endoderm and creates a 'sloppy spliced' nuclear transcript.

## *T-REX17* does not regulate SOX17

To investigate the functional role of *T-REX17* during endoderm formation, we first generated a cell line carrying a constitutive transcriptional repressor (dCas9-KRAB-MeCP2, *Yeo et al., 2018*). We

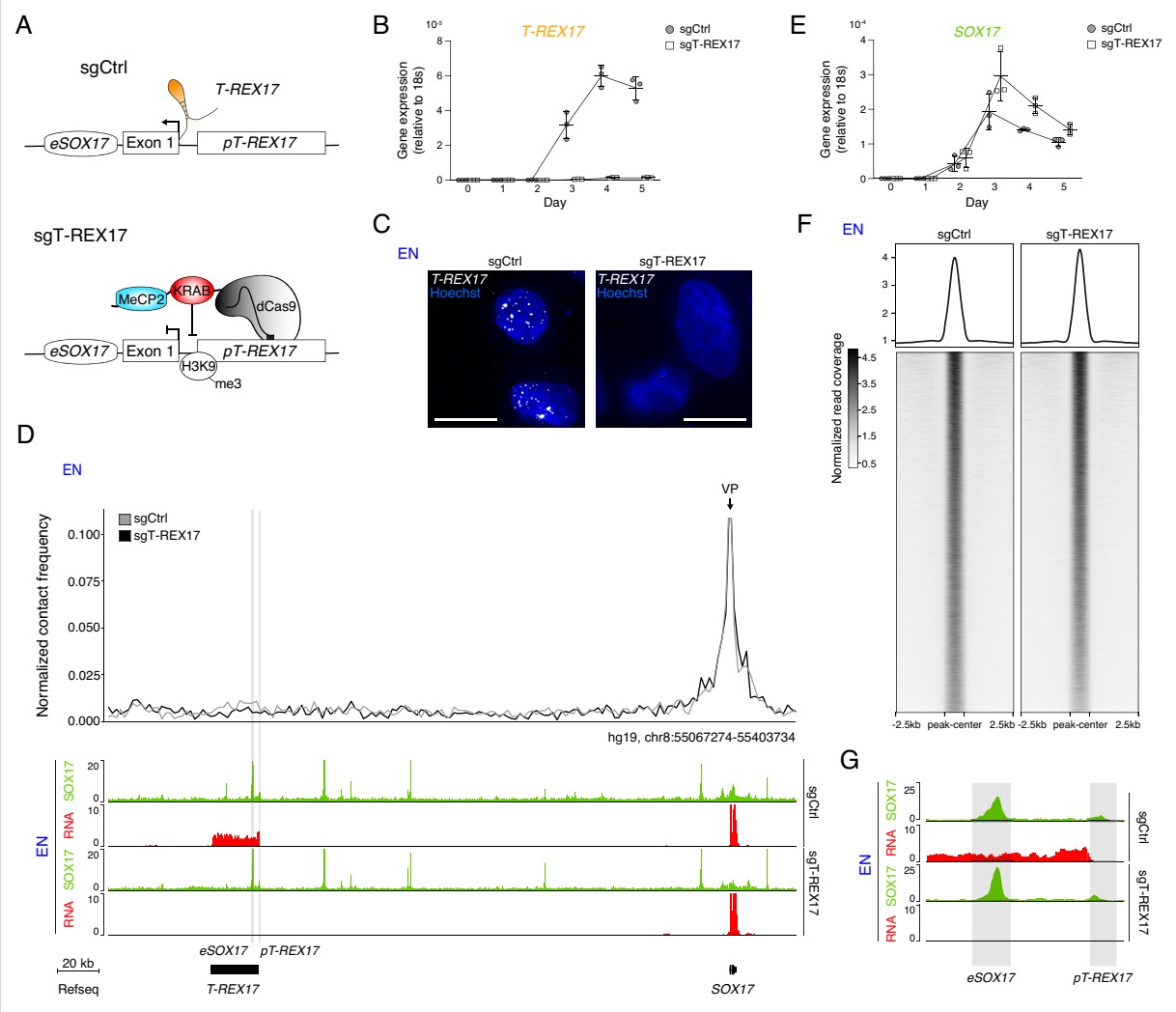

**Figure 3.** *T-REX17* regulation at the *SOX17* topological domain. (**A**) Schematic of *T-REX17* locus regulation in the absence (top) or presence (bottom) of a targeting dCas9-KRAB-MeCP2 complex, decorating *T-REX17* promoter with an H3K9me3 mark 355 bp upstream of the TSS. (**B**) Time-resolved qRT-PCR showing the expression of *T-REX17* during EN differentiation in the presence or absence of dCas9-KRAB-MeCP2 complex targeting *T-REX17* promoter (normalized to the housekeeping gene *18s*). Symbols indicate the mean and error bars indicate SD across three independent experiments. Individual data points are displayed. (**C**) smRNA-FISH of *T-REX17* in sgCtrl (left) and sgT-REX17 (right) EN cells counter-stained with Hoechst. Scale bars, 10 µm. For an extended field of view see **Figure 3—figure supplement 1B**. (**D**) Virtual 4C analysis from capture Hi-C experiments in sgCtrl and sgT-REX17 EN cells using *SOX17* promoter as viewpoint, with 2 kb resolution (upper panel). SOX17 EN ChIP-seq (RPKM) and RNA-seq (CPM) profiles in the two conditions are shown in the tracks (lower panel). *eSOX17* and *pT-REX17* are highlighted in grey. (**E**) Time-resolved qRT-PCR showing the expression of *SOX17* during EN differentiation in the presence or absence of dCas9-KRAB-MeCP2 complex targeting *T-REX17* promoter (normalized to the housekeeping gene *18s*). Symbols indicate the mean and error bars indicate SD across three independent experiments. Individual data points are displayed. (**F**) Heatmap showing SOX17 binding distribution genome-wide in sgCtrl and sgT-REX17 EN. The displayed peaks represent the union of the identified peaks in the two conditions (n=61.153). (**G**) SOX17 ChIP-seq and RNA-seq tracks at the *T-REX17* locus showing SOX17 binding at the SOX17 enhancer (*eSOX17*) and *T-REX17* promoter (*pT-REX17*). SOX17 binding on *pT-REX17* results in *T-REX17* activation, if *pT-REX17* is not targeted by dCas9-KRAB-MeCP2.

The online version of this article includes the following source data and figure supplement(s) for figure 3:

**Figure supplement 1.** SOX17 and *T-REX17* reciprocal gene expression regulation.

**Figure supplement 1—source data 1.** Source data for the genotyping gels and blot in **Figure 3—figure supplement 1G, I, M**.

**Figure supplement 2.** *T-REX17* interacts with HNRNPU.

**Figure supplement 2—source data 1.** Source data for the blot in **Figure 3—figure supplement 2E**.

then derived two cell lines from it, one harboring a control sgRNA (sgCtrl) designed by a randomization approach of human TSS regions (*Gilbert et al., 2014*) and the other specifically targeting the *T-REX17* promoter (sgT-REX17; see Materials and methods) (*Figure 3A*). Immunofluorescent staining for dCas9-KRAB-MeCP2 demonstrated its homogeneous expression in the parental cell line (*Figure 3—figure supplement 1A*). The dCas9-mediated silencing resulted in a strong repression of *T-REX17* RNA compared to the control, which we further validated by smRNA-FISH (*Figure 3B and C*; *Figure 3—figure supplement 1B*). We furthermore detected H3K9me3 enrichment around the *T-REX17* promoter in sgT-REX17 cells, with a certain degree of spreading toward the enhancer *eSOX17* but no apparent consequence on *SOX17* regulation (*Figure 3—figure supplement 1C*). To assess possible effects of the *T-REX17* depletion on *SOX17*, we performed Capture Hi-C (cHi-C) in both cell lines, but could not observe any significant interaction differences (Log$_2$FC = 0.02 p=0.049) within the *SOX17*-loop domain in definitive endoderm (*Figure 3—figure supplement 1D*). Nevertheless, virtual 4C analysis revealed a marginal decrease in the *SOX17* enhancer-promoter interaction in the absence of *T-REX17* (*Figure 3D*). Despite this limited topological difference, loss of *T-REX17* does not appear to affect *SOX17* transcriptional activation and expression levels, indicating preserved enhancer functionality (*Figure 3D and E*). We also confirmed that unrelated genes present in neighboring domains were unaffected by the perturbation (*Figure 3—figure supplement 1E*).

Next, we performed SOX17 Chromatin Immunoprecipitation sequencing (ChIP-seq) and show that SOX17 occupancy at the *SOX17* locus (including at its induced heterochromatic distal enhancer (*eSOX17*)) as well as genome-wide is largely unaffected by the loss of *T-REX17* (*Figure 3D and F*; *Figure 3—figure supplement 1F*). Interestingly, we found SOX17 enrichment at the *T-REX17* promoter (*pT-REX17*), potentially contributing to its activation and consistent with the timing relative to SOX17 (*Figures 3D, G , and 2A*). To further explore this relationship we generated heterozygous (*SOX17^{WT/Δ}*) and homozygous (*SOX17^{Δ/Δ}*) *SOX17* knock-out cell lines (*Figure 3—figure supplement 1G–I*). Notably, homozygous knock-out cells fail to induce the expression of the endoderm master regulator *GATA4*, and show no activation of *T-REX17* (*Figure 3—figure supplement 1J*).

In order to distinguish between the function of *T-REX17* active transcription and its actual transcript (*Allou and Balzano, 2021*; *Daneshvar et al., 2016*), we generated an additional cell line by introducing a strong transcriptional termination signal downstream of an mRuby cassette into the first exon of *T-REX17*, hereafter T-REX17^{p(A)/p(A)} (*Figure 3—figure supplement 1K–M*). qRT-PCR demonstrated that the expression of *T-REX17* is abolished in T-REX17^{p(A)/p(A)} EN cells, while the mRuby cassette is actively transcribed, indicating ongoing transcription at the locus in an endoderm-specific manner (*Figure 3—figure supplement 1N*). In line with our depletion experiments, *SOX17* expression levels are not affected in T-REX17^{p(A)/p(A)} EN cells (*Figure 3—figure supplement 1N*).

These results demonstrate that *T-REX17* induction is dependent on SOX17, whereas the *T-REX17* transcript and the act of transcription are dispensable for SOX17 activation as well as its genome-wide localization.

## *T-REX17* interacts with HNRNPU

To explore how *T-REX17* is involved in endoderm regulation, we investigated whether it was associated with RNA binding proteins, a common way lncRNAs exert their functions (*Hudson et al., 2014*; *Xue et al., 2016*; *Duszczyk et al., 2011*; *Brown et al., 2014*; *Chillón and Pyle, 2016*). To this end, we performed RNA-pulldown followed by mass spectrometry (*Figure 3—figure supplement 2A–B*). Among the putative *T-REX17* interactors, we identified several heterogenous nuclear ribonucleoprotein (hnRNP) family members, including HNRNPU (*Figure 3—figure supplement 2C*). HNRNPU waspreviously reported to interact with lncRNAs to regulate various functions during development including nuclear matrix organization (*Hacisuleyman et al., 2014*; *Alvarez-Dominguez et al., 2017*), X chromosome inactivation (*Hasegawa et al., 2010*), RNA splicing (*Xiao et al., 2012*; *Huelga et al., 2012*), and epigenetic control of gene expression (*Khyzha et al., 2019*; *Song et al., 2020*; *Puvvula et al., 2014*). To validate HNRNPU-*T-REX17* interaction, we performed HNRNPU RNA immunoprecipitation (RIP) (*Figure 3—figure supplement 2D and E*) and found *T-REX17* to be enriched to levels comparable to known RNA interactors such as *XIST* or *NEAT1* (*Figure 3—figure supplement 2F*).

Although more work is required, our preliminary analysis identified known lncRNA-interacting ribonucleoproteins that may help resolve the molecular function of *T-REX17*.

## *T-REX17* is required for the differentiation toward definitive endoderm

To investigate the cellular role of *T-REX17*, we performed immunofluorescent staining and fluorescent activated cell sorting (FACS) for CXCR4 in control and *T-REX17*-depleted cells. The latter showed a substantial reduction in the CXCR4[+] cell population during differentiation, suggesting hampered differentiation potential toward endoderm (*Figure 4A*). However, consistent with the transcriptional data, SOX17 protein levels were not affected (*Figure 4A*). Both phenotypes were recapitulated in the T-REX17[p(A)/p(A)] EN cells (*Figure 4—figure supplement 1A*). As expected, based on its highly restricted expression, differentiation toward the other two germ layers (mesoderm and ectoderm) was not affected (*Figure 2—figure supplement 1D*; *Figure 4—figure supplement 1B,D*).

Next, we performed time-resolved RNA-seq in *T-REX17* depleted and control cell lines on days 0, 3, and 5 of endoderm differentiation. Principal Component Analysis (PCA) revealed only marginal variance by day 3, while a more substantial transcriptional divergence was observed on day 5 (*Figure 4—figure supplement 1D*). Differential gene expression analysis identified 584 significantly down- and 590 significantly upregulated genes in *T-REX17*-depleted cells at day 5 (*Figure 4B*). In particular, we found pluripotency genes (e.g. *POU5F1*, *NANOG*) and endoderm/WNT-related genes (e.g. *EOMES*, *GATA3*, *CXCR4*, *FZD5*, *FZD7*, *FZD8*, *DKK1*, *NOTUM*, *ROR1*, *CXXC4*, *SFRP5*) to be significantly up- and downregulated, respectively (*Figure 4B*; *Figure 4—figure supplement 1E*). Time resolved qPCR analysis over 5 days confirmed, a lack of key endoderm markers activation and expression in *T-REX17*-depleted cells (including *CXCR4*, *GATA3*, *GATA4*, *KLF5*, *CPE*, *GPR*, *HHEX*, *EPSTI1*, *FOXA3*), an aberrant transcriptional signature we also observe in T-REX17[p(A)/p(A)] EN cells (*Martinez Barbera et al., 2000*; *Grapin-Botton and Constam, 2007*; *McLean et al., 2007*; *Séguin et al., 2008*; *Teo et al., 2011*; *Aksoy et al., 2014*; *Dettmer et al., 2020*; *Figure 4—figure supplement 1F–H*). Interestingly, among the significantly, upregulated genes in *T-REX17*-depleted cells, we found an enrichment of *JUN* (AP-1) pathway target genes (including *EGR1*, *ATF3*, *PVR*, *DAB2*, *NOTCH2*, *MFHAS1*, *SPARC*) (*Briggs et al., 2002*; *Schummer et al., 2016*; *Florin et al., 2004*; *van Dam and Castellazzi, 2001*; *Hoffmann et al., 2008*; *Kockel et al., 2001*), which has recently been described to act as a barrier for the exit from pluripotency toward endoderm formation (*Figure 4B*; *Figure 4—figure supplement 1E*; *Li et al., 2019*). Phosphorylation levels of JUN-activating upstream kinase JNK are a strong indicator of JUN pathway activation (*Raivich and Behrens, 2006*; *Muniyappa and Das, 2008*; *Li et al., 2019*), which we observed by increased relative amounts of pJNK in *T-REX17*-depleted cells (*Figure 4C*; *Figure 4—figure supplement 2A*). Inhibition of JNK hyperactivity (JNK Inhibitor XVI) from day 3 of definitive endoderm differentiation partially rescued the specification defect in *T-REX17*-depleted cells (*Figure 4—figure supplement 2B and C*).

Furthermore, immunofluorescent staining for ECAD, NCAD, and VIM revealed retention of an epithelial signature in *T-REX17* depleted endoderm cells (*Figure 4D and E*; *Figure 4—figure supplement 1E*; *Figure 4—figure supplement 2D and E*). Moreover, VIM-signal distribution within *T-REX17*-depleted cells was also altered, indicating a potential cellular polarization defect (*Figure 4E*; *Figure 4—figure supplement 2E*).

Finally, we evaluated if *T-REX17*-depleted cells have lost the potential to further differentiate into pancreatic progenitor (PP) cells (*Alvarez-Dominguez et al., 2020*). Immunofluorescent staining identified a very distinct PDX1[+] population in the control cell population after 9 days of directed differentiation, which is notably reduced in *T-REX17*-depleted cells (*Figure 4F and G*; *Figure 4—figure supplement 2F*). In addition, transcriptomic analysis of differentiated control and *T-REX17*-depleted cells indicates a substantial gene expression difference, including the specific downregulation of pancreatic progenitor marker genes (*Alvarez-Dominguez et al., 2020*; *Figure 4H*; *Figure 4—figure supplement 2G*).

Our data therefore highlight the importance of *T-REX17* for the induction of definitive endoderm, which directly impacts the subsequent differentiation potential.

## Discussion

Here, we describe the discovery and characterization of *T-REX17* as a functionally essential lncRNA in human definitive endoderm. Most lncRNAs act locally, regulating the chromatin architecture and the expression of neighboring genes in *cis-* (*Tan et al., 2017*; *Wang et al., 2011*; *Goff et al., 2015*; *Engreitz et al., 2016*), especially when overlapping with enhancer elements. In particular, the

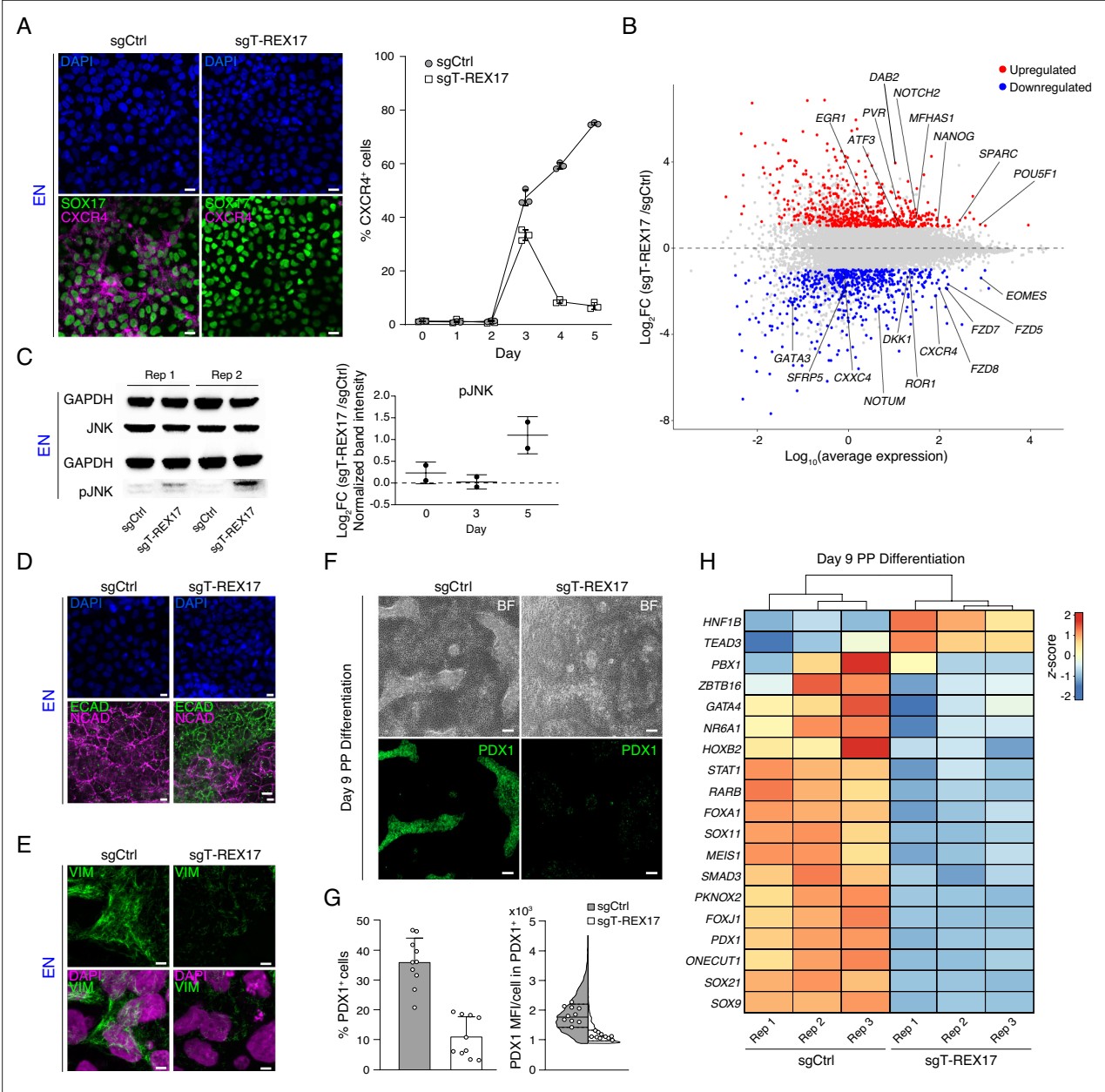

**Figure 4.** Endodermal defects in cells depleted of *T-REX17*. (**A**) Immunofluorescent (IF) staining of SOX17 and CXCR4 in EN cells expressing either sgCtrl or sgT-REX17 counter-stained with DAPI (left panel). Line plot showing percentage of FACS-derived CXCR4⁺ cell population at given time points during endoderm differentiation (right panel). Symbols indicate mean values, while error bars show SD across three independent experiments. Individual data points are displayed. Scale bars, 10 µm. (**B**) Scatter plot highlighting differentially expressed genes between sgT-REX17 and sgCtrl EN cells. Significantly (Log₂FC ≥1, p-value <0.05) upregulated genes (n=590 upon *T-REX17* repression are shown in red while significantly (Log₂FC ≤ –1, p-value <0.05) down-regulated genes (n=584) are shown in blue. Three independent replicates have been performed. The complete lists of TPMs and differentially expressed genes are provided in *Supplementary file 2*. (**C**) JNK and pJNK western blots of sgCtrl and sgT-REX17 EN cells (left panel). GAPDH signals are used as loading controls above the corresponding JNK/pJNK signals. Boxplot showing relative pJNK levels during endoderm differentiation. Quantification is depicted as Log₂FC of sgT-REX17 over sgCtrl (right panel) and provided in *Supplementary file 1*. Central line indicates the mean, error bars indicate the SD across two independent experiments. Differentiation time-course blots are shown in *Figure 4—figure supplement 2A*. (**D**) IF staining of ECAD and NCAD in EN cells expressing either sgCtrl or sgT-REX17 counter-stained with DAPI. Scale bars, 10 µm. (**E**) IF staining of VIM in EN cells expressing either sgCtrl or sgT-REX17 counter-stained with DAPI. Scale bars, 5 µm. (**F**) Bright field images of PP differentiation cultures (upper panel) followed by IF staining for PDX1 (lower panel) of either sgCtrl or sgT-REX17 cells. Scale bars, 10 µm. (**G**) IF staining quantification of overall (sgCtrl, n=17.657, sgT-REX17, n=5.279 analyzed cells) PDX1⁺ population percentages (left) or PDX1 mean fluorescence intensity distribution in PDX1⁺ cells (right). Bar plot error bars indicate SD around the mean value and white dots represent mean values for the individual replicates (N=10). Lines of the violin plot indicate interquartile range around the median value and white dots represent median values for the individual replicates (N=10). List

*Figure 4 continued on next page*

*Figure 4 continued*

of values for each cell and corresponding statistics are shown in **Supplementary file 4**. (**H**) Heatmap showing row-normalized *z*-scores of PP specific marker genes (**Alvarez-Dominguez et al., 2020**) in sgCtrl and sgT-REX17 EN cells as measured by RNA-seq at day 9 of differentiation. Columns were ordered by hierarchical clustering (represented as tree above the heatmap). Note the reduced expression of PP master transcription factor *PDX1* in sgT-REX17 as compared to sgCtrl. The complete lists of TPMs and differentially expressed genes are provided in **Supplementary file 2**.

The online version of this article includes the following source data and figure supplement(s) for figure 4:

**Source data 1.** Source data for the blot in *Figure 4C*.

**Figure supplement 1.** Molecular phenotypes associated with the loss of *T-REX17*.

**Figure supplement 2.** Cellular phenotypes associated with the loss of *T-REX17*.

**Figure supplement 2—source data 1.** Source data for the blots in *Figure 4—figure supplement 2A, B*.

---

fine-tuned expression of several developmental transcription factors has been shown to rely on the activity of lncRNAs present within the same topological domain (**Wang et al., 2011**; **Frank et al., 2019**; **Messemaker et al., 2018**). Interestingly, *T-REX17* appears distinct from these and other endodermal specific lncRNAs (**Jiang et al., 2015**; **Liao et al., 2019**; **Daneshvar et al., 2016**) as it does not appear to regulate the adjacent *SOX17* gene. The use of two orthogonal loss of function approaches in our work (suppression of *T-REX17* activation and early termination) showed that *T-REX17* transcription is dispensable for proper *SOX17* regulation. It remains to be determined what the targets and regulatory mechanism of *T-REX17* are. One may speculate that these could be distant and unrelated loci to the *SOX17* loop-domain, as we find many *T-REX17* distinct puncta in the nuclear compartment of endodermal cells. Typically, local *cis*-acting lncRNAs mainly show accumulation at the two sites of nascent transcription (**Jiang et al., 2015**; **Lewandowski et al., 2019**; **Daneshvar et al., 2016**; **Daneshvar et al., 2020**). The observed interaction with the HNRNP complex may link it to various nuclear-related functions needed for endoderm specification. It will be interesting to see how this compares to other endodermal lncRNAs, which mainly exert their functions together with endoderm-specific transcription factors (**Jiang et al., 2015**; **Daneshvar et al., 2020**; **Chen et al., 2020**; **Yang et al., 2020**).

SOX genes are fundamental transcription factors that have a variety of functions including the specification of cell types and tissues during embryonic development. They are evolutionary conserved and evolved as a result of a series of ancient genomic duplication events (**Bowles et al., 2000**). Interestingly, at other SOX gene loci, the presence of one or multiple lncRNAs have been reported, but these lncRNAs, in contrast to *T-REX17*, appear involved in the modulation of the associated SOX gene expression in *cis-* (**Tariq et al., 2020**; **Barter et al., 2017**; **Amaral et al., 2009**; **Ahmad et al., 2017**). This suggests that lncRNAs near paralogous genes may evolve distinct role and regulatory mechanisms.

At a functional level, our results show that *T-REX17* is essential for definitive endoderm specification and its loss limits further downstream differentiation, as demonstrated by the pancreatic progenitor differentiation. How the different phenotypic changes associated with the loss of *T-REX17* arise, such as an aberrant endodermal transcriptome, EMT-failure, JNK-hyperactivity and lack of pancreatic progeny, remains unclear. Advanced biochemical assays to simultaneously profile RNA-RNA, RNA-DNA, and RNA-protein interactions (**Chu et al., 2012**; **Quinodoz et al., 2018**; **Engreitz et al., 2015**) might help elucidating the mechanism of action by which *T-REX17* controls endodermal transition. From the developmental perspective, *T-REX17* and its transient, highly stage-specific nature make it an intriguing regulator compared to most of the protein-coding genes, including endodermal transcription factors, for example SOX17, FOXA2, and GATA4, which are expressed longer and in a variety of somatic tissues. In this context, it is worth noting that the development of definitive endoderm during human gastrulation in vivo takes place within hours and the gene regulatory network (GRN) governing this transition has to be tightly controlled (**Tsankov et al., 2015**; **Gifford et al., 2013**; **Chia et al., 2019**), which could also involve lncRNAs such as *T-REX17*.

As such, our study contributes toward a more complete understanding of the multi-layered regulation of human cellular differentiation and connects it to a previously unannotated non-coding RNA.

## Materials and methods

Default parameters were used, if not otherwise specified, for all software and pipelines utilized in this study.

### Molecular cloning of *SOX17* and e*SOX17.2* knock-out constructs

For CRISPR/Cas9 mediated targeting of either *SOX17* (Addgene plasmid #195494) or *eSOX17.2* (Addgene plasmid #195495) we utilized our previously generated two small guide RNAs (sgRNAs) at once expression system 2 X_pX458_pSpCas9(BB)–2A-GFP (Addgene plasmid #172221). sgRNA-cloning was performed with NEBuilder HiFi DNA Assembly Master Mix (New England Biolabs, E2621S) according to manufacturer's instructions using BbsI-linearization of 2 X_pX458 for the first sgRNA and SapI linearization of 2 X_pX458 for the second sgRNA as backbone, combined with single stranded oligonucleotides containing the sgRNA sequences as inserts (1:3 molar ratio; find sequence in *Supplementary file 3*). Bacterial transformation and Sanger sequencing were performed to verify successful cloning.

### Molecular cloning of Luciferase reporter constructs

pGL4.27[luc2P/minP/Hygro] (Promega, E8451) containing a minimal CMV-promoter for enhancer-assays or pGL4.15[luc2P/Hygro] (Promega, E6701) w/o any promoter for promoter-assays were first digested using EcoRV (New England Biolabs, R3195S). Next, full *eSOX17* (Addgene plasmid # 195498), *eSOX17.1* or *eSOX17.2* (Addgene plasmid # 195499) for enhancer-assays and *pSOX17* (Addgene plasmid # 195496) or *pT-REX17* (Addgene plasmid # 195497) genomic regions were PCR amplified with primers containing homology overhangs to the plasmid. PCR products were purified and cloned into the linearized plasmid utilizing the NEBuilder HiFi DNA Assembly Master Mix (1:3 molar ratio) according to the manufacturer's instructions. Bacterial transformation followed by Sanger sequencing verified the successful cloning. Cloning primers are listed in *Supplementary file 3*.

### Molecular cloning of lentiviral sgRNA constructs

pU6-sgRNA EF1Alpha-puro-T2A-BFP (*Gilbert et al., 2014*) was digested with BstXI (New England Biolabs, R0113S) and BlpI (New England Biolabs, R0585S) and the linearized plasmid was gel extracted with the QIAquick Gel Extraction Kit (Quiagen, 28704). Subsequently sgRNA containing oligonu-cleotides (sgT-REX17 or sgCtrl) (s. *Supplementary file 3*) were cloned in the linearized backbone using NEBuilder HiFi DNA Assembly Master Mix (1:3 molar ratio) according to the manufacturer's instructions to finally obtain pU6-sgT-REX17_EF1a-Puro-T2A-BFP (Addgene plasmid #195501) and pU6-sgCtrl_EF1a-Puro-T2A-BFP (Addgene plasmid #195500). Bacterial transformation and sanger sequencing confirmed the successful cloning. pU6-sgRNA EF1Alpha-puro-T2A-BFP (*Gilbert et al., 2014*) was a gift from Jonathan Weissman (Addgene plasmid # 60955; http://n2t.net/addgene:60955; RRID:Addgene_60955).

### Molecular cloning of *SOX17* reporter knock-in constructs

pUC19 plasmid was digested with SmaI (New England Biolabs, R0141S) and the linearized plasmid was gel extracted with the QIAquick Gel Extraction Kit (Quiagen, 28704). Next, *SOX17* homology arm genomic regions were PCR amplified with primers containing homology overhangs to the plasmid and to a T2A-H2B-mCitrine-loxP-hPGK-BSD-loxP selection cassette.

The left homology arm overlapped with the end of the *SOX17* coding sequence, and the T2A-H2B-mCitrine cassette which was cloned in frame with the last *SOX17* aminoacid. PCR products and selection cassette were purified and cloned into the linearized pUC19 to finally generate pUC19_T2A-H2B-mCitrine_loxP-hPGK-BSD-loxP (Addgene plasmid #195503) utilizing the NEBuilder HiFi DNA Assembly Master Mix according to the manufacturer's instructions. Bacterial transformation followed by Sanger sequencing verified the successful cloning.

sgRNA targeting the genomic region of integration (SOX17 C-terminus) was cloned in BbsI linearized pX335-U6-Chimeric_BB-CBh-hSpCas9n(D10A) (*Cong et al., 2013*) plasmid (Addgene plasmid #42335) to finally generate pX335_U6-Chimeric_BB-CBh-hSpCas9n(D10A)_SOX17_C-term_KI (Addgene plasmid #195502) using NEBuilder HiFi DNA Assembly Master Mix (1:3 molar ratio) according to the manufacturer's instructions. pX335-U6-Chimeric_BB-CBh-hSpCas9n(D10A) was a gift from Feng Zhang (Addgene plasmid # 42335; http://n2t.net/addgene:42335; RRID:Addgene_42335).

Bacterial transformation and sanger sequencing confirmed the successful cloning. Cloning primers are listed in *Supplementary file 3*.

## Molecular cloning of *T-REX17*-promoter-KI constructs

pUC19 plasmid was digested with SmaI (New England Biolabs, R0141S) and the linearized plasmid was gel extracted with the QIAquick Gel Extraction Kit (Quiagen, 28704). Next, *T-REX17* homology arm genomic regions were PCR amplified with primers containing homology overhangs to the plasmid and to a mRuby-3xFLAG-NLS-3xSV40-poly(A)_loxP-mPGK-PuroR-loxP selection cassette to finally generate pUC19_mRuby-3xFLAG-NLS-3xSV40-poly(A)_loxP-mPGK-PuroR-loxP (Addgene plasmid #195505).

The left homology arm overlapped with the *T-REX17* promoter including 30 bp of *T-REX17* Exon 1, and a mRuby-3xFLAG-NLS-3xSV40-poly(A) cassette which was cloned +30 bp after *T-REX17*-TSS into Exon 1. The right homology arm overlapped with *T-REX17* Exon 1–30 bp TSS, and a loxP-mPGK-PuroR-loxP cassette, which was cloned following the mRuby-3xFLAG-NLS-3xSV40-poly(A) cassette, originating from a synthetic oligonucleotide (GenScript Biotech). Both the mRuby-3xFLAG-NLS-3xSV40-poly(A) and the loxP-mPGK-PuroR-loxP cassette also shared homology. All PCR products were purified and cloned into the linearized plasmid utilizing the NEBuilder HiFi DNA Assembly Master Mix according to the manufacturer's instructions. Bacterial transformation and Sanger sequencing verified the successful cloning.

For Cas9 mediated targeting of the *T-REX17* promoter we utilized pSpCas9(BB)–2A-Puro (PX459) V2.0 (*Ran et al., 2013*), which was a gift from Feng Zhang (Addgene plasmid # 62988; http://n2t. net/addgene:62988; RRID:Addgene_62988) (*Ran et al., 2013*). sgRNA-cloning was performed with NEBuilder HiFi DNA Assembly Master Mix (New England Biolabs, E2621S) according to manufacturer's instructions using BbsI-linearization of PX459, combined with single stranded oligonucleotides containing the sgRNA sequences as inserts (1:3 molar ratio) (find sequence in *Supplementary file 3*) to finally obtain pX459_V2.0_pSpCas9(BB)–2A-Puro_T-REX17_Ex1_KI (Addgene plasmid #195504). Bacterial transformation and Sanger sequencing were performed to verify successful cloning.

## hiPS cell culture

ZIP13K2 (*Tandon et al., 2018*) hiPSCs were maintained in mTeSR1 (Stemcell Technologies, 85850) on pre-coated culture ware (1:100 diluted Matrigel (Corning, 354234) in KnockOut DMEM (Thermo Fisher Scientific, 10829–018)). Clump-based cell splitting was performed by incubating the cells in final 5 mM EDTA pH 8,0 (Thermo Fisher Scientific, 15575–038) in DPBS (Thermo Fisher Scientific, 14190250) 5 min at 37 °C, 5% $CO_2$. Single-cell splitting was performed by incubating the cells with Accutase (Sigma-Aldrich, A6964) supplemented with 10 µM Y-27632 (Tocris, 1254) for 15 min at 37 °C, 5% $CO_2$. Cell counting was performed using a 1:1 diluted single-cell suspensions in 0,4% Trypan Blue staining-solution (Thermo Fisher Scientific, 15250061) on the Countess II automated cell-counter (Thermo Fisher Scientific). Wash-steps were performed by spinning cell-suspensions at 300 x g 5 min at room temperature (RT).

## Definitive endoderm (EN) differentiation

To guarantee high reproducibility, constant media-quality, and mTeSR1 compatibility, definitive endoderm differentiations were exclusively performed utilizing the STEMdiff Trilineage Endoderm Differentiation media (Stemcell Technologies, 05230). Single-cell suspensions of mTeSR1 maintained ZIP13K2 hiPSCs were seeded into the respective culture formats according to the required cell-number as recommended by the manufacturer's instructions. Media change using the STEMdiff Trilineage Endoderm Differentiation media was performed on a daily bases according to the manufacturer's instructions. Cells were then collected at required timepoints by washing the plate with DPBS before single-cell dissociation was performed with Accutase for 15 min at 37 °C, 5% $CO_2$. Single-cell suspensions of definitive endoderm (EN) differentiated cells were utilized for further downstream analysis (qPCR, western blot, FACS etc.).

## Embryoid body (EB) formation followed by ScoreCard Assay

ZIP13K2 hiPSC single cell suspensions were prepared and counted as previously described (s. *hiPS cell culture*). Next, 1x10³ cells/well of either sgCtrl or sgT-REX17 hiPSCs were seeded on a 96-well ultra-low attachment U-bottom plate (Corning, 7007) in respective cell culture media.

## Random EB differentiation

Cells were seeded in 200 µl /well of hES-media (Final DMEM-F12 (Thermo Fisher Scientific, 11320074), 20% KSR (Thermo Fisher Scientific, 10828028), 1% Penicillin /Streptomycin, 1% NEAA (Thermo Fisher Scientific, 11140050), 0,5% GlutaMAX, HEPES (Thermo Fisher Scientific, 31330038)), supplemented with final 10 µM Y-27632. Single-cell suspensions were spun at 100 x g for 1 min at RT and further cultured for 16 hr at 37 °C, 5% $CO_2$. The following day 150 µl media supernatant was carefully exchanged by 150 µl fresh hES-media (without Y-27632). Cells were further cultured for additional 48 hr at 37 °C, 5% $CO_2$. The very same media was replaced every 48 hr until day 9. At day 9, EBs were collected, washed once in DPBS and RNA isolated (s. *RNA isolation and cDNA synthesis*).

## Undifferentiated control EBs

Cells were seeded in 200 µl /well of mTeSR1, supplemented with final 10 µM Y-27632. Single-cell suspensions were spun at 100 x g for 1 min at RT and further cultured for 16 hr at 37 °C, 5% $CO_2$. The following day 150 µl media supernatant was carefully exchanged by 150 µl fresh mTeSR1 media (without Y-27632). Cells were further cultured for additional 48 hr 37 °C, 5% $CO_2$. At day 3, EBs were collected, washed once in DPBS and RNA isolated (s. *RNA isolation and cDNA synthesis*).

cDNA-conversion and ScoreCard assay (Thermo Fisher Scientific, A15870) has been performed according to the manufacturer's instructions.

## JNK inhibition experiments

For the JNK-inhibition experiments, 1 µM JNK inhibitor XVI (Sellekchem, S4901) final was supplemented to the media from day 3 of EN differentiation onward. The corresponding volume of DMSO was supplemented to the media of the control samples.

## Pancreatic progenitor (PP) differentiation

Pancreatic progenitor (PP) differentiation was performed as previously described (*Alvarez-Dominguez et al., 2020*) with minor changes. Briefly, single-cell suspensions of ZIP13K2 hiPSCs (s. *hiPS cell culture*) were seeded at a density of 5x10⁵ cells /cm² in mTeSR1 supplemented with 10 µM Y-27632. After 24 hr, culture medium was replaced with S1-media (Final 11.6 g/L MCDB131, Sigma Aldrich, M8537-1L; 2 mM D-+-Glucose, Sigma Aldrich, G7528-250G; 2.46 g/L NaHCO3, Sigma Aldrich, S5761-500G; 2% FAF-BSA, Proliant Biologicals, 68700–1; 1:50,000 of 100 x ITS-X, Thermo Fisher Scientific, 51500056; 1 x GlutaMAX, Thermo Fisher Scientific, 35050–038; 0.25 mM ViatminC, Sigma-Aldrich, A4544-100G; 1% Pen-Strep, Thermo Fisher Scientific, 15140122) supplemented with final 100 ng/ml Activin-A (R&D Systems, 338-AC-01M) and 1.4 µg/ml CHIR99021 (Stemgent, 04-0004-10). The following 2 days, cells were cultured in S1-media supplemented with final 100 ng/ml Activin-A. Next, cells were cultured in S2-media (Final 11.6 g/L MCDB131; 2 mM D-+-Glucose; 1.23 g/L NaHCO3; 2% FAF-BSA; 1:50,000 of 100 x ITS-X; 1 x GlutaMAX; 0,25 mM ViatminC; 1% Pen-Strep) supplemented with final 50 ng/ml KGF (Peprotech, 100-19-1MG) for 48 hr. After these 48 hr, cells were cultured in S3-media (Final 11.6 g/L MCDB131; 2 mM D-+-Glucose; 1.23 g/L NaHCO3; 2% FAF-BSA; 1:200 of 100 x ITS-X; 1 x GlutaMAX; 0.25 mM ViatminC; 1% Pen-Strep) supplemented with final 50 ng/ml KGF (Peprotech, 100-19-1MG), 200 nM LDN193189 (Sigma Aldrich, SML0559-5MG), 0.25 µM Sant-1 (Sigma Aldrich, S4572-5MG), 2 µM Retinoic Acid (Sigma Aldrich, R2625-50MG), 500 nM PDBU (Merck Millipore, 524390–5 MG) and 10 µM Y-27632 for 24 hr. Finally, cells were cultured in the previous S3-media composition w/o supplementation of LDN193189 for 24 hr. Between daily media changes, cells were washed once with 1 x DPBS. Throughout the entire differentiation process, cells were cultured at 37 °C, 5% $CO_2$ in 100 µl media /cm².

## Luciferase reporter assays

ZIP13K2 hiPSCs (s. *hiPS cell culture*) were treated with Accutase containing 10 µM Y-27632 for 15 min, 37 °C, 5% $CO_2$ to obtain a single cell suspension. Cell suspensions were counted and seeded at a

density of $10^5$ cells /$cm^2$ in mTeSR1 supplemented with final 10 µM Y-27632. Sixteen hours later, cells were co-transfected with 15 fmol pRL-TK (Promega, E2241) and 150 fmol of either pGL4.27[luc2P/ minP/Hygro] empty vector or pGL4.27[luc2P/minP/Hygro] containing either *eSOX17*, *eSOX17.1* or *eSOX17.2* utilizing Lipofectamin Stem Transfection Reagent (Thermo Fisher Scientific, STEM00003) following the manufacturer's instructions. Transfection was performed in mTeSR1 containing 10 µM Y-27632 for 16 hr at 37 °C, 5% $CO_2$. Subsequently, endoderm differentiation was initiated (day 0) using the STEMdiff Trilineage Endoderm Differentiation media. At days 0, 2, 3, or 5 of endoderm differentiation, cells were lysed and Renilla as well as Firefly Luciferase activity was measured using the Dual-Glo Luciferase Assay System (Promega, E2920) according to the manufacturer's instructions. Raw values (*Supplementary file 1*) were measured on the GloMax-Multi Detection System (Promega).

## Generation of *SOX17* and *eSOX17.2* CRISPR/Cas9 knock-out hiPSC lines

ZIP13K2 hiPSCs (s. *hiPS cell culture*) were treated with Accutase containing final 10 µM Y-27632 for 15 min at 37 °C, 5% $CO_2$ to obtain a single cell suspension. Cell suspensions were counted and seeded at a density of 1–2 x $10^5$ cells /$cm^2$ in mTeSR1 supplemented with final 10 µM Y-27632. Cells were pre-cultured for 16 hr at 37 °C, 5% $CO_2$ prior to transfection.

Cells were then transfected with 6 µg /6-well of P2X458 using Lipofectamin Stem Transfection Reagent according to the manufacturer's instructions. GFP$^+$ cells were FACS-sorted 16–24 hr post-transection with the FACSAria II or the FACSAria Fusion (Beckton Dickinson) and seeded at a density of 0,5–1 x $10^3$ cells /$cm^2$ in mTeSR1 supplemented with 10 µM Y-27632 to derive isogenic clones. Single-cell derived colonies were manually picked, and split half for maintenance in a well of a 96-well plate and half used for genotyping using the Phire Animal Tissue Direct PCR Kit (Thermo Fisher Scientific, F140WH) following manufacturer's instructions. Genotyping primer are listed in *Supplementary file 3*. Edited alleles were verified by cloning PCR-products into the pJET1.2 backbone (Thermo Fisher Scientific, K1232) according to the manufacturer's instructions, followed by bacterial transformation and sanger sequencing.

## Generation of *SOX17*-reporter hiPS cell line

ZIP13K2 hiPSCs (s. *hiPS cell culture*) were treated with Accutase containing final 10 µM Y-27632 for 15 min at 37 °C, 5% $CO_2$ to obtain a single cell suspension. Cell suspensions were counted and seeded at a density of 1–2 x $10^5$ cells /$cm^2$ in mTeSR1 supplemented with final 10 µM Y-27632. Cells were pre-cultured for 16 hr at 37 °C, 5% $CO_2$ prior to transfection.

The following day, cells were transfected using Lipofectamin Stem Transfection Reagent in fresh mTeSR1 supplemented with final 10 µM Y-27632 for 24 hr at 37 °C, 5% $CO_2$. Transfection mixtures contained 3 µg of T2A-H2B-mCitrine-loxP-hPGK-BSD-loxP donor plasmid and 3 µg of PX335-SOX17 (1:1 molar ratio) per 6-well.

Two days post transfection, cells were selected with final 2 µg/ml Blasticidin-S-HCl (Thermo Fisher Scientific, A1113903) for 14 days at 37 °C, 5% $CO_2$. For the derivation of isogenic reporter cell lines, single-cell derived colonies were manually picked and expanded. Differentiation into EN followed by FACS analysis was used to confirm clones that were activating the reporter.

## Generation of *T-REX17*-promoter-KI hiPS cell line

ZIP13K2 SOX17-reporter (s. *Generation of SOX17-reporter hiPS cell line*) hiPSCs (s. *hiPS cell culture*) were treated with Accutase containing final 10 µM Y-27632 for 15 min at 37 °C, 5% $CO_2$ to obtain a single cell suspension. Cell suspensions were counted and seeded at a density of 1–2 x $10^5$ cells /$cm^2$ in mTeSR1 supplemented with final 10 µM Y-27632. Cells were pre-cultured for 16 hr at 37 °C, 5% $CO_2$ prior to transfection.

The following day, cells were transfected using Lipofectamin Stem Transfection Reagent in fresh mTeSR1 supplemented with final 10 µM Y-27632 for 24 hr at 37 °C, 5% $CO_2$. Transfection mixtures contained 3 µg of mRuby-3xFLAG-NLS-3xSV40-poly(A)-loxP-mPGK-PuroR-loxP donor plasmid and 3 µg of PX458-T-REX17-promoter (1:1 molar ratio) per 6-well.

Two days post transfection, cells were selected with final 2 µg/ml Puromycin-Dihydrochloride (Thermo Fisher Scientific, A1113803) for 14 days at 37 °C, 5% $CO_2$. For the derivation of isogenic

reporter cell lines, single-cell derived colonies were manually picked and expanded. Differentiation into EN followed by qRT-PCR analysis was used to confirm clones that were activating the reporter.

## Generation of dCas9-KRAB-MeCP2 hiPS cell line

ZIP13K2 hiPSCs (s. *hiPS cell culture*) were treated with Accutase containing final 10 µM Y-27632 for 15 min at 37 °C, 5% $CO_2$ to obtain a single cell suspension. Cell suspensions were counted and seeded at a density of 1–2 x $10^5$ cells /$cm^2$ in mTeSR1 supplemented with final 10 µM Y-27632. Cells were pre-cultured for 16 hr at 37 °C, 5% $CO_2$ prior to transfection.

The following day, cells were transfected using Lipofectamin Stem Transfection Reagent in fresh mTeSR1 supplemented with final 10 µM Y-27632 for 24 hr at 37 °C, 5% $CO_2$. Transfection mixtures contained 2 µg of Super PiggyBac transposase expression vector (SBI, PB210PA-1) and 4 µg dCas9-KRAB-MeCP2 (*Yeo et al., 2018*) (1:1 molar ratio) per 6-well. dCas9-KRAB-MeCP2 was a gift from Alejandro Chavez & George Church (Addgene plasmid # 110821; http://n2t.net/addgene:110821; RRID:Addgene_110821).

Two days post transfection, cells were selected with final 2 µg/ml Blasticidin-S-HCl (Thermo Fisher Scientific, A1113903) for 14 days at 37 °C, 5% $CO_2$. For the derivation of isogenic CRISPRi cell lines, single-cell derived colonies were manually picked and expanded. IF stainings for Cas9 confirmed homogenous dCas9-KRAB-MeCP2 expression in the selected clones (s. *Immunofluorescence staining* for detailed experimental procedure).

## Production of lentiviral particles carrying sgRNAs

Lentiviral particles of specific sgRNA constructs have been produced in HEK-293T cells by co-transfection of 1:1:1 molar ratios pCMV-VSV-G plasmid (addgene, #8454 *Stewart et al., 2003*, 3,5 µg), psPAX2 plasmid (addgene, #12260, 7 µg) in combination with sgRNA-specific variants of pU6-sgRNA EF1Alpha-puro-T2A-BFP (*Gilbert et al., 2014*) plasmid (addgene, #60955, 14 µg). pCMV-VSV-G was a gift from Bob Weinberg (Addgene plasmid # 8454; http://n2t.net/addgene:8454; RRID: Addgene_8454). Prior to transfection, HEK-293T cells were grown on a 10 cm dish up to 70–80% confluency in HEK-media (KO-DMEM (Themro Fisher Scientific, 10829018), 10% fetal bovine serum (FBS, PAN Biotech, P30-2602), 1 x GlutaMAX Supplement, 100 U/ml Penicillin-Streptomycin (Thermo Fisher Scientific, 15140122) and final 1 x, 5,5 µM ß-Mercaptoethanol (Thermo Fisher Scientific, 21985023)). For each sgRNA construct, plasmid DNA mixtures and 50 µl of LipoD293 transfection reagent (SignaGen Laboratories, SL100668) were mixed in 250 µl KO-DMEM at RT. After pipette mixing, transfection particles were incubated at RT for 15 min. Each sgRNA-specific mixture was added drop-wise onto HEK-293T cultures in 10 ml HEK-media and incubated for 16 hr at 37 °C, 5% $CO_2$. Cell culture media was exchanged by 10 ml fresh HEK-media the next day and culture supernatants (S/N) of the two subsequent days were then filtered (0.22 µm), collected and stored at 4 °C. After the second harvesting day, S/N were supplemented with 1 x PEG-it virus precipitation solution (SBI, LV810A-1) for 24 h at 4 °C. Viral particles were finally precipitated by centrifugation at 3234 x g, 4 °C. Viral precipitates were resuspended in 200 µl mTeSR1 and either frozen at –80 °C or immediately used for lentiviral transduction of CRISPRi hiPSCs. The entire lentivirus preparation and storage was carried out under S2-safety conditions and precautions.

## Lentiviral transduction of dCas9-KRAB-MeCP2 hiPSCs

Lentiviral particles were either thawed on ice (if frozen) or directly used fresh on the day of production. For hiPS cells transduction, clump-based hiPSCs splitting was performed (s. *hiPS cell culture* for detailed experimental procedure) and dissociated clumps were supplemented with 10 µM Y-27632, 10 µg/ml Polybrene infection reagent (MerckMillipore, TR-1003-G) and 100 µl lentiviral particles preparation. Cells were then plated and cultured for 16 hr at 37 °C, 5% $CO_2$. The following day, cells were washed 10 times with DPBS and given fresh mTeSR1 supplemented with 10 µM Y-27632 for 24 hr at 37 °C, 5% $CO_2$.

Successfully infected cells were then selected with 2 µg/ml Puromycin Dihydrochloride (Thermo Fisher Scientific, A1113803) for 14 days at 37 °C, 5% $CO_2$. dCas9-KRAB-MeCP2 cell lines expressing sgRNAs (sgT-REX17 and sgCtrl), were grown as bulk cultures, and Tag-BFP was used as a proxy for sgRNA expression prior to differentiation into the respective endodermal derivate.

## RNA isolation and cDNA synthesis

For RNA extraction, cells were lysed in 500 µl Qiazol from the miRNeasy Mini Kit (Quiagen, 217004), followed by vortexing. RNA was then extracted using the miRNeasy Mini Kit (Quiagen, 217004) and RNA concentration was measured. cDNA synthesis was performed using 1 µg total RNA for each sample using the RevertAid First Strand cDNA Synthesis Kit (Thermo Fisher Scientifc, K1622), following the manufacturer's instructions Random hexamers have been used as primers for first strand cDNA synthesis.

## Quantitative PCR (qPCR)

Quantitative PCR (qPCR) was carried out on a StepOnePlus 96-well or a QuantStudio 7 Flex 384-well Real-Time PCR System (Thermo Fisher Scientific) loading 20–25 ng cDNA /well and using TaqMan Fast Advanced Master-Mix (Thermo Fisher Scirentific, 4444557) with TaqMan validated probes (*Supplementary file 3*) (Thermo Fisher Scientific) following the manufacturer's instructions.

## 5'/3' RACE PCR experiments

5'/3' rapid amplification of cDNA ends (RACE) PCR reactions where performed utilizing the 5'/3' RACE Kit, second generation (Sigma-Aldrich, 3353621001) according to the manufacturer's instructions. Corresponding gene specific (SP) primers are listed in *Supplementary file 3*.

RACE-PCR products were cloned into pJET1.2 backbone followed by bacterial transformation and sanger sequencing.

## Extraction of polyA RNA for Nanopore sequencing

Isolation of poly(A)-enriched mRNA was performed using the Dynabeads mRNA DIRECT purification kit (Thermo Fisher Scientific, 61011) according to the manufacturer's instruction with minor modifications. ZIP13K2-derived EN cells were washed once with DPBS and dissociated with Accutase for 15 min at 37 °C, 5% $CO_2$. Enzymatic reaction was quenched by adding mTeSR1 and cells were counted using the Countess II automated cell-counter. A total of $4 \times 10^6$ viable cells were centrifuged for 5 min at 4 °C, 300 x g. The supernatant was discarded and cells were washed with 1 ml of ice-cold DPBS and centrifuged as described above. The supernatant was completely removed and the cell pellet was carefully resuspended in 1.25 ml Lysis/Binding buffer. In order to reduce viscosity resulting from released genomic DNA, the samples were passed through a 21 gauge needle (Becton Dickinson, 304432) for five times and subsequently added to the pre-washed Oligo(dT)$_{25}$ beads. Hybridization of the beads/mRNA complex was carried out for 10 min on a Mini Rotator (Grant-bio) and vials were placed on a DynaMag2 magnet (Thermo Fisher Scientific, 12321D) until the beads were fully immobilized. The DNA containing supernatant was removed and the beads were resuspended twice with 2 ml of Buffer A following a second wash step with two times 1 ml of Buffer B. Purified RNA was eluted with 10 µl of pre-heated Elution Buffer (10 mM Tris-HCl pH 7,5) for 5 min at 80 °C and quantified with a Qubit Fluorometer (Thermo Fisher Scientific) using the RNA HS Assay Kit (Thermo Fisher Scientific, Q32852). Eluted RNA samples were immediately used for preparation of Nanopore sequencing libraries or kept at –80 °C.

## Preparation of Nanopore sequencing libraries

Preparation of RNA sequencing libraries was performed following the manufacturer's instructions (ONT, SQK-PCS109) with minor modifications. Briefly, 50 ng of freshly prepared poly(A)-enriched mRNA was subjected to reverse transcription and strand-switching reaction. A total of four PCR reactions, each containing 5 µl of reverse transcribed cDNA, was used for the attachment of rapid primers (cPRM). Sufficient amplification of long cDNA molecules was enabled by setting the PCR extension time to 19 min and a total of 12 x cycles were used for amplification. Samples were treated with 1 µl of Exonuclease I (New England Biolabs, M0293S) and subsequently pooled for SPRI bead cleanup. Wash steps were performed using 80% ethanol solution and beads were eluted in 60 µl of 50 °C pre-heated nuclease-free water. Samples were then incubated for additional 20 min at 50 °C. Eluted DNA was combined with 5 µl adapter mix (AMX), 25 µl ligation buffer (LNB) from ONTs ligation sequencing kit (ONT, SQK-LSK109) and 10 µl of NEBNext Quick T4 DNA Ligase (New England Biolabs, E6056S). Ligation mix was incubated at RT for 30 min. Removal of short DNA fragments was achieved by adding 40 µl of Agencourt AMPure XP beads (Beckmann Coulter, A63881) combined with two wash

steps with 250 µl of long fragment buffer (LFB) included in ONTs ligation sequencing kit. The final library was eluted in 13 µl elution buffer (EB) for 20 min at 48 °C and DNA concentration was quantified using the Qubit dsDNA BR assay kit (Thermo Fisher Scientific, Q32850). A total of 400 ng was carefully mixed with 37.5 µl sequencing buffer (SQB), 25.5 µl of loading beads (LB) and loaded onto a primed MinION flow cell (ONT, R9.4.1 FLO-MIN106).

## RNA sequencing

ZIP13K2 hiPSCs and their derived EN cultures were treated with Accutase for 15 min at 37 °C, 5% $CO_2$ to obtain a single cell suspension. Cells were then collected, washed with ice cold DPBS and centrifuged at 4 °C, 300 x g for 5 min. Subsequently, 350 µl of RLT Plus buffer containing 1% β-mercaptoethanol (Thermo) was added to the cell pellets for cell lysis. After dissociation by trituration and vortexing, RNA was extracted using RNeasy Plus Micro Kit (Qiagen) and RNA concentration and quality was measured using the Agilent RNA 6000 Pico Kit (Agilent Technologies, 5067–1513) on an Agilent 2100 Bioanalyzer. All samples analyzed had a RINe value higher than 8.0, and were subsequently used for library preparation. mRNA libraries were prepared using KAPA Stranded RNA-Seq Kit (KapaBiosystem) according to the manufacturer's instructions. A total of 500 ng of total RNA was used for each sample to enter the library preparation protocol. For adapter ligation dual indexes were used (NEXTFLEX Unique Dual Index Barcodes NOVA-514150) at a working concentration of 71 nM (5 µl of 1 uM stock in each 70 µl ligation reaction). Quality and concentration of the obtained libraries were measured using Agilent High Sensitivity D5000 ScreenTape (Agilent-Technologies, 5067–5592) on an Agilent 4150 TapeStation. All libraries were sequenced using 100 bp paired-end sequencing (200 cycles kit) on a NovaSeq platform at a minimum of 25 million fragments /sample.

## 4C sequencing

Triplicates of either undifferentiated ZIP13K2 or ZIP13K2-derived EN cultures were collected as described previously. ZIP13K2-derived EN cultures were further quenched with MACS-buffer (Final DPBS, 2 mM EDTA (ThermoFisher Scientific), 0.5% BSA (Sigma-Aldrich)) to obtain a single cell suspension. CXCR4+ cell populations, were enriched using MicroBead Kit (Miltenyi Biotec) following the manufacturer's instructions. Pre- and post-MACS enriched cell fractions of differentiated cultures were measured for CXCR4-APC signal on the FACS Aria II (Beckton Dickinson) to confirm the cell population purity. Circularized Chromosome Conformation Capture (4 C) library preparation of undifferentiated, or differentiated CXCR4+ enriched cell populations was performed according to the Weintraub A.S. et al. protocol (*Weintraub et al., 2017*). Briefly, NlaIII (New England Biolabs, R0125) was used as the primary cutter and DpnII (New England Biolabs, R0543) as a secondary cutter. Touchdown PCR on 4 C libraries was performed using specific primer-pairs (s. primer list in *Supplementary file 3*) for the respective view-points. Illumina sequencing libraries were then prepared and sequenced using 150 paired-end sequencing (300 cycles kit) on a HiSeq4000 platform at a minimum of 10 M fragments/ sample.

## Capture Hi-C sequencing

cHi-C libraries were prepared from CRISPRi sgCtrl or sgT-REX17 EN cells. 5x10⁶ ZIP13K2-derived EN cells were harvested and washed with ice cold DPBS. Cell lysis, NlaIII (NEB, R0125) digestion and proximity-ligation was performed according to the Franke et al. protocol (*Franke et al., 2016*) with minor changes. Adaptors were added to DNA fragments and amplified according to Agilent Technologies instructions for Illumina sequencing. The library was hybridized to the custom-designed SureSelect probes (Agilent Technologies, 5190–4806/3253271) (s. probe list in *Supplementary file 3*) and indexed for sequencing of 200 M fragments /sample (100 bp paired-end) following the Agilent instructions. Capture Hi-C experiments were performed as biological duplicates.

## SOX17 chromatin immunoprecipitation (ChIP) sequencing

ZIP13K2-derived EN cells (5x10⁶ / IP) were harvested and cross-linked in 1% formaldehyde (Thermo Fisher Scientific, 28908) in DPBS for 10 min at RT, followed by quenching with final 125 mM Glycine (Sigma-Aldrich, 50046) for 5 min at RT. Cross-linked cells were then centrifuged at 500 x g at 4 °C and washed twice with ice cold DPBS. Cell lysis was performed by resuspending the pellet in 500 µl Cell Lysis Buffer (Final 10 mM Tris-HCl, pH 8,0 (Sigma Aldrich, T2694); 85 mM KCl (Sigma Aldrich, P9541);

0,5% NP40 (Sigma Aldrich, 56741); 1 x cOmplete, EDTA-free Protease Inhibitor Cocktail (Sigma Aldrich, 11873580001)) followed by 10 min incubation on ice. After the incubation, lysed cells were centrifuged at 2500 x g for 5 min at 4 °C. Supernatant was carefully removed and the extracted nuclei were then resuspended in 230 µl Nuclei Lysis Buffer (Final 10 mM Tris-HCl, pH 7,5 Sigma Aldrich, T2319); 1% NP40; 0.5% sodium deoxycholate (Sigma Aldrich, D6750); 0,1% SDS (Thermo Fisher Scientific, AM9820); 1 x cOmplete, EDTA-free Protease Inhibitor Cocktail. Following 10 min incubation on ice, each 260 µl sample was split into two microTUBEs (Covaris, 520045) and chromatin was sonicated using a Covaris E220 Evolution with the following settings: Temperature → 4 °C; Peak power → 140; Duty factor → 5,0; Cycles/Burst → 200; Duration → 750 sec. After sonication, sheared chromatin (ranging from 200 to 600 bp) was transferred in a new 1.5 ml tube and centrifuged at max speed for 10 min at 4 °C. Supernatant was then transferred into a new tube and volume was increased to 1 ml /sample with ChIP Dilution Buffer (Final 16.7 mM Tris-HCl, pH 8.0; 1.2 mM EDTA Sigma Aldrich, 03690); 167 mM NaCl (Sigma Aldrich); 1,1% Triton-X (Sigma Aldrich); 0.01% SDS; 1 x Protease Inhibitor. Fifty µl (5%) was then transferred into a new tube and frozen at –20 °C as INPUT. One µg of SOX17 antibody /$10^6$ initial cells was added to the 950 µl left, and immunoprecipitation was carried out at 4 °C o/n on a rotator (*Supplementary file 3*). The next day, 50 µl of Dynabeads Protein G (Thermo Fisher Scientific, 10004D) /IP were washed twice with ice cold ChIP Dilution Buffer and then added to each IPs. IP/bead mixes were incubated for 4 hr at 4 °C on a rotor. Next, bead/chromatin complexes were washed twice with Low Salt Wash Buffer at 4 °C (Final 20 mM Tris-HCl, pH 8,0; 2 mM EDTA; 150 mM NaCl (Sigma-Aldrich, S6546); 1% Triton-X; 0,1% SDS), twice with High Salt Wash Buffer at 4 °C (Final 20 mM Tris-HCl, pH 8.0; 2 mM EDTA; 500 mM NaCl; 1% Triton-X; 0.1% SDS), twice with LiCl Wash Buffer at 4 °C (Final 10 mM Tris-HCl, pH 8.0; 1 mM EDTA; 250 mM LiCl (Sigma Aldrich, L9650); 1% sodium deoxycholate (Sigma Aldrich); 1% NP40), twice with TE pH 8.0 (Sigma Aldrich, 8890) at room temperature and finally eluted twice in 50 µl freshly prepared ChIP Elution Buffer (Final 0,5% SDS; 100 mM NaHCO3 (Sigma Aldrich, S5761)) at 65 °C for 15 min (total 100 µl final eluent). Thawed INPUTS and eluted IPs were next reverse cross-linked at 65 °C o/n after the addition of 16 µl freshly prepared Reverse Crosslinking Salt Mixture (Final 250 mM Tris-HCl, pH 6,5 (Sigma Aldrich, 20–160); 62.5 mM EDTA; 1,25 M NaCl; 5 mg/ml Proteinase K (Thermo Fisher Scientific, AM2548)). The following day, phenol:chloroform (Thermo Fisher Scientific, 15593031) extraction followed by precipitation was performed to isolate DNA. IPs and INPUTS were then quantified and NGS libraries were prepared using NEBNext Ultra II DNA Library Prep Kit for Illumina (New England Biolabs, #E7645) following the manufacturer's instructions. Library quality and size distribution was verified using a TapeStation D5000 HS kit (Agilent Technologies, 5067–5592). Samples were sequenced with a coverage of 50 M paired end reads (2x100 bp) /sample on a NovaSeq (Illumina).

## GATA4/GATA6 chromatin immunoprecipitation (ChIP) sequencing

GATA4/6 ChIPs were perfored in duplicates as previously described (*Genga et al., 2019*). Briefly, approximately 5x106 cells were used for each IP. Cells were cross-linked with 1% formaldehyde for 10 min followed by quenching with 125 mM glycine for 4–5 min at room temperature. The cell pellet was lysed in cell lysis buffer (20 mM Tris-HCl pH 8, 85 mM KCl, 0.5% NP-40) supplemented with 1 X protease inhibitors (Roche, 11836170001) on ice for 20 min then spun at 5000 rpm for 10 min. The nuclear pellet was resuspended in sonication buffer (10 mM Tris pH 7.5, 1% NP-40, 0.5% sodium deoxycholate, 0.1% SDS, and 1 X protease inhibitors) and incubated for 10 minutes at 4 °C. In order to achieve a 200–700 bp DNA fragmentation range, nuclei were sonicated using a Bronson sonifier (model 250) with the following conditions: amplitude = 15%, time interval = 3 min (total of 8–12 min) and pulse ON/OFF = 0.7 s/1.3 s. Chromatin was pre-cleared with Dynabeads Protein A (Invitrogen, 10002D) for 1 hr and incubated with antibody on a rotating wheel overnight at 4 °C. On the following day, 30–40 µl of Dynabeads Protein A was added to chromatin for 2–3 hr. The captured immuno-complexes were washed as follows – 1 x in low-salt buffer, 1 x in high-salt buffer, 1 x in LiCl salt buffer, and 1 x in TE. The immuno-complexes were eluted in ChIP-DNA elution buffer (10 mM Tris-HCl pH 8, 100 mM NaCl, 20 mM EDTA, and 1% SDS) for 20 min. The eluted ChIP-DNA was reverse cross-linked overnight at 65 °C, followed by proteinase K (Thermo, 25530049) treatment, RNase A (Thermo, ENO531) treatment, and Phenol:Chloroform:Isoamyl alcohol extraction. The Illumina library construction steps were carried out with 5–10 ng of purified DNA. During library construction, purification was performed after every step using QIAquick PCR purification kit (QIAGEN, 28104) or QIAquick

gel extraction kit (QIAGEN, 28706). The library reaction steps were as follows: end-repair, 3′ end A-base addition, adaptor ligation, and PCR amplification. The amplified libraries were size-selected for 200–450 bp on a 2% agarose E-gel (Thermo, G402002) and sequenced (single-end, 75) on a NextSeq500 or Hi-Seq2000 platform.

## H3K9me3 chromatin immunoprecipitation (ChIP) qPCR

ZIP13K2-derived EN cells ($2x10^6$ / IP) were harvested, cross-linked, washed, lysed, and sonicated as described previously (s. *SOX17 ChIP sequencing*). ChIP for H3K9me3 was performed in triplicates utilizing the High-Sensitivity ChIP Kit (abcam, ab185913) in combination with the ChIP-grade H3K9me3 antibody (ab8898, abcam) according to the manufacturer's instructions with slight modifications. Instead of DNA column purification, phenol:chloroform extraction followed by precipitation was performed to isolate DNA (s. *SOX17 ChIP sequencing*). Precipitated DNA was dissolved in 200 µl $H_2O$.

qPCR reactions were set up utilizing the 2 x PowerUp SYBR Green Master Mix (Thermo Fisher Scientific, A25777) containing final 250 nM forward /reverse primer (s. *Supplementary file 3*). All samples have been measured in technical triplicates using 4 µl diluted input or IP sample from above /reaction /replicate. qPCRs were set-up on 96-well plates (Thermo Fisher Scientific, N8010560), spun down for 1 min at 2500 x g, RT and ran on a StepOnePlus 96-well Real-Time PCR System (Thermo Fisher Scientific).

## *T-REX17* RNA-pulldown followed by mass spectrometry

RNA-pulldown protocol to discover *T-REX17* protein interaction partners has been performed combining (*Engreitz et al., 2014*; *Chu et al., 2012*) protocols with some modifications. ZIP13K2-derived EN cells ($60x10^6$) were harvested and cross-linked in 1% formaldehyde (Thermo Fisher Scientific, 28908) in DPBS for 5 min at RT, followed by quenching with final 125 mM Glycine (Sigma-Aldrich, 50046) for 5 min at RT. Cross-linked cells were then centrifuged at 500 x g at 4 °C and washed three times with ice cold DPBS. Cells are then resuspended in 10 ml Sucrose/Glycerol buffer (1:1) (*Sucrose Buffer*: 0.3 M Sucrose; 1% Triton-X (Sigma Aldrich); 10 mM HEPES (Thermo Fisher Scientific, 31330038); 100 mM KOAc; 0.1 mM EGTA (Sigma Aldrich); 0.5 mM Spermidine; 0.15 mM Spermine; 1 mM DTT; 1 X proteinase inhibitor (Roche, 11836170001); 10 U/ml SUPER-asIN (Thermo Fisher Scientific, AM2694)) (*Glycerol Buffer*: 25% Glycerol; 10 mM HEPES; 100 mM KOAc; 0.1 mM EGTA; 1 mM EDTA (Sigma Aldrich, 03690); 0.5 mM Spermidine; 0.15 mM Spermine; 1 mM DTT; 1 X proteinase inhibitor; 10 U/ml SUPER-asIN) and dounced 20 times in a glass tight pestle (Sigma Aldrich, D9938-1SET). After douncing, lysed cells are incubated for 10 min on ice inside the pestle. Cells are then transferred on a cushion of 10 ml Glycerol Buffer in a 50 ml falcon tube and centrifuged at 1000 x g for 15 min at 4 °C to recover nuclei. Supernatant is discarded by pipetting first, and residual volume is decanted on a clean paper towel. Extracted nuclei are then resuspended in 5 ml 3% formaldehyde and fixed again for 30 min at RT, followed by three DPBS washes. Next, nuclei are resuspended in 5 ml Nuclei Extraction Buffer (Final 50 mM HEPES, pH 7,5; 250 mM NaCl; 0,1% sodium deoxycholate (Sigma Aldrich, D6750); 0,1 mM EGTA; 0,5% N-lauroylsarcosine; 5 mM DTT; 100 U/ml SUPER-asIN) and incubated for 10 min on ice. Nuclei are then centrifuged at 400 x g for 5 min at 4 °C, and resuspended in 530 µl Nuclei Resuspension Buffer (Final 50 mM HEPES, pH 7.5; 75 mM NaCl; 0.1% sodium deoxycholate; 0.1 mM EGTA; 0.5% N-lauroylsarcosine; 5 mM DTT; 100 U/ml SUPER-asIN) and sonicated using a Covaris E220 Evolution with the following settings: Temperature → 4 °C; Peak power → 140; Duty factor → 5,0; Cycles/Burst → 200; Duration → 15 min. After sonication, sheared chromatin is split into 3 samples (Even/Odd/LacZ, 120 µl each) and incubated with the corresponding biotinylated probes set (36 pmols of probes are added; see *Supplementary file 3* for probes sequences) together with 240 µ. Hybridization Buffer (Final 33 mM HEPES, pH 7.5; 808 mM NaCl; 0,33% SDS; 5 mM EDTA; 0.17% N-lauroylsarcosine; 2.5 mM DTT; 5 X Denhardt's solution; 1 X proteinase inhibitor; 100 U/ml SUPER-asIN) overnight at RT on a rotor. 5% sonicated sample was frozen as INPUT. The next day, 240 µl of MyOne Streptavidin C1 beads (Thermo Fisher Scientific, 65001) were added to each pulldown after washing and resuspension in Hybridization Buffer, and incubated for 3 hr at RT on a rotor. Next, bead complexes were washed once with Wash Buffer 1 (Final 30 mM HEPES, pH 7.5; 1.5 mM EDTA; 240 mM NaCl; 0.75% N-lauroyl-sarcosine; 0.65% SDS; 0,7 mM EGTA; 2 M Urea), four times with Wash Buffer 2 (Final 10 mM HEPES, pH 7.5; 2 mM EDTA; 240 mM NaCl; 0,1% N-lauroylsarcosine; 0.2% SDS; 1 mM EGTA) and once with

RNase H elution Buffer (Final 50 mM HEPES, pH 7.5; 1.5 mM EDTA; 75 mM NaCl; 0.125% N-lauroylsarcosine; 0.5% Triton-X; 10 mM DTT; 0.5 M Urea). In this last step, 10% of the beads from each pulldown is transferred to a new tube for RNA isolation. The remaining 90% (protein sample fraction) is eluted in RNase H elution Buffer containing 10% RNase H, 10% RNase A and 10% DNase for 30 min at RT. The RNase fraction is de-crosslinked together with the INPUT samples with Proteinase K (Thermo Fisher Scientific, AM2548) treatment and RNA is extracted following Trizol purification. RNA and INPUT samples were reverse transcribed and used for qPCR to validate *T-REX17* enrichment. Protein samples were run on a NuPAGE 4–12%, Bis-Tris, 1.0 mm, Mini Protein Gel, Silver stained using SilverQuest (Thermo Fisher Scientific; LC6070) following manufacturer instructions. The mass spectrometry compatible SilverQuest Silver Staining Kit was used for de-staining. Gel pieces were then washed twice with 300 µL of 25 mM ammonium bicarbonate in 50% acetonitrile, shaking at 500 rpm for 10 min, followed by centrifugation at 16,000 x g for 30 s. Gel pieces were completely dried in a vacuum concentrator. In-gel digestion with trypsin and extraction of peptides was done as previously described (*Kaiser et al., 2008*). Dried peptides were reconstituted in 5% acetonitrile and 2% formic acid in water, briefly vortexed, and sonicated in a water bath for 30 s before injection to nano-LC-MS. LC-MS/MS was carried out by nanoflow reverse-phase liquid chromatography (Dionex Ultimate 3000, Thermo Scientific) coupled online to a Q-Exactive HF Orbitrap mass spectrometer (Thermo Scientific), as reported previously (*Gielisch and Meierhofer, 2015*). Briefly, the LC separation was performed using a PicoFrit analytical column (75 µm ID ×50 cm long, 15 µm Tip ID; New Objectives, Woburn, MA) in-house packed with 3 µm C18 resin (Reprosil-AQ Pur, Dr. Maisch, Ammerbuch, Germany). Peptides were eluted using a gradient from 3.8 to 38% solvent B in solvent A over 120 min at a 266 nL/min flow rate. Solvent A was 0.1% formic acid and solvent B was 79.9% acetonitrile, 20% $H_2O$, and 0.1% formic acid. For the IP samples, a 1-hr gradient was used. Nanoelectrospray was generated by applying 3.5kV. A cycle of one full Fourier transformation scan mass spectrum (300–1750 m/z, resolution of 60,000 at m/z 200, automatic gain control (AGC) target $1 \times 10^6$) was followed by 12 data-dependent MS/MS scans (resolution of 30,000, AGC target $5 \times 10^5$) with a normalized collision energy of 25 eV.

## HNRNPU RNA immunoprecipitation (RIP) followed by qRT-PCR or western blot

ZIP13K2-derived EN cells ($10 \times 10^6$) were harvested and cross-linked according to the manufacturer's instructions in 0.3% formaldehyde in DPBS for 10 min at RT, followed by quenching with final 1 x Glycine solution for 5 min at RT utilizing the Magna Nuclear RIP (Cross-Linked) Nuclear RNA-Binding Protein Immunoprecipitation Kit (Merck millipore, 17–10520). Cross-linked cells were then centrifuged at 800 x g at 4 °C and washed three times with ice cold DPBS. Supernatant free cell pellets were conducted to cell lysis according to the Kit manufacturer's instructions. Sonication has been performed in Kit provided RIP Cross-linked Lysis Buffer using the Covaris E220 Evolution with the following settings: Temperature → 4 °C; Peak power → 140; Duty factor → 5,0; Cycles/Burst → 200; Duration → 6 min. to obtain a DNA smear of 200–1000 bp. Sonicated lysates were centrifuged at 1000 x g for 10 min at 4 °C and supernatants aliquoted and stored at –80 °C. DNase I treatment following Immunoprecipitation has been performed according to the Kit manufacturer's instructions, combining lysates corresponding to $10^6$ cells with 5 µg antibody per sample (*Supplementary file 3* for antibodies). After DNase I treatment 10% input material for qRT-PCR has been kept and stored at –80 °C. Initial supernatants (unbound fraction w/o beads) after o/n immunoprecipitation and 10% material of the last wash step (IP including beads) has been kept for Western Blot and stored at –20 °C. Inputs and IP were further conducted to reverse crosslinking and RNA purification according to the Kit manufacturer's instructions. cDNA synthesis has been carried out as mentioned earlier (s. *RNA isolation and cDNA synthesis*).

Quantitative PCR (qPCR) reactions were set up utilizing the 2 x PowerUp SYBR Green Master Mix (Thermo Fisher Scientific, A25777) containing final 250 nM forward /reverse primer (s. *Supplementary file 3* for primer & oligos) and 20–25 ng cDNA /well. Reactions were set up in 384-well plates (Thermo Fisher Scientific, AB2384B) following centrifugation for 2 min at 2500 x g, RT. Reactions were carried out on a QuantStudio 7 Flex 384-well Real-Time PCR System (Thermo Fisher Scientific).

Western Blot samples of unbound fractions and IP were boiled in final 1 x Laemmli Buffer (BioRad, 1610747) containing 10% 2-Mercaptoethanol (M6250, Sigma-Aldrich) for 10 min at 95 °C, followed by

cooling on ice for 5 min. Western blots have finally been carried out as described below (s. *Western Blot*) utilizing respective antibody dilutions (s. *Supplementary file 3* for antibodies).

## Immunofluorescence staining

For immunofluorescent stainings, cells were grown in Ibidi eight-well glass-bottom plates (Ibidi, 80827) (initial seeding, $10^4$ cells /well). On the day of analysis, cells were washed twice with DPBS and then fixed in 4% Paraformaldehyde (PFA) solution (Sigma-Aldrich, P6148-500G) for 30 min at 4 °C, and then washed three more times with DPBS. Subsequently, cells were permeabilized for 30 min in DPBS-T solution (Final 0.5% Triton-X (Sigma-Aldrich, T8787-50 ML) in DPBS) and blocked for 30 min in Blocking solution (Final 10% fetal bovine serum in DPBS-T) at RT. Primary antibody incubation was performed in blocking solution for 1 hr and 45 min at RT, after which cells were washed three times with Blocking solution. After the last washing step, samples were incubated with secondary antibodies diluted in Blocking solution for 30 min at RT. Afterwards, cells were washed three times with DPBS-T. The last DPBS-T washing step after secondary antibody incubation contained 0.02% DAPI (Roche Diagnostics, 10236276001). DAPI was incubated for 10 min at RT and washed off once with DPBS. All primary and secondary antibodies and their working concentrations are listed in *Supplementary file 3*.

## Cell clearing

Prior to imaging, cells were cleared with RIMS (Refractive Index Matching Solution) in order to increase light penetrability. To this end, samples were first washed three times with 0.1 M phosphate buffer (0.025 M $NaH_2PO_4$, 0.075 M $Na_2HPO_4$, pH 7.4). Clearing was then performed by incubation in RIMS solution (133% w/v Histodenz (Sigma-Aldrich, D2158) in 0.02 M phosphate buffer) at 4 °C o/n.

## Immunofluorescence imaging

Cells stained with antibodies were imaged with the Zeiss Celldiscoverer7 (wide-field), Zeiss LSM880 (laser-scanning microscope with Airyscan), Zeiss Observer (wide-field) or Nikon Eclipse TS2 (bench-top microscope) with appropriate filters for DAPI, Alexa Fluor 488, Alexa Fluor 568, Alexa Fluor 647, and combinations thereof.

## Quantitative fluorescence microscopy

For each staining tested, a total of 49 individual positions were acquired in 3 fluorescence channels /replicate /well, with a 20 x /NA = 0.95 objective, an afocal magnification changer 1 x, 3x3 camera binning, a consequential pixel size of 0.46 $\mu m^2$, and in constant focus stabilization mode. Analysis was then performed using the Image Analysis module running in ZEN 3.2. On average 6928 single cells were analyzed per replicate. Cells were identified on smoothed nuclear counterstaining (DAPI) using fixed intensity thresholds, nearby objects were separated by mild water shedding. The consequential primary objects were filtered (area 45–175 $\mu m^2$) and expanded by 8 pixels (=5.44 $\mu m^2$); the consecutive ring, surrogated a cytoplasm compartment. Fluorescence intensities (mean and standard deviation) were quantified for each nucleus and expanded object, depending on the staining pattern profiled.

## Single-molecule RNA fluorescent in situ hybridization

For single-molecule RNA fluorescent in situ hybridization (smRNA-FISH), cells were grown in Ibidi eight-well glass-bottom plates (Ibidi 80827) (initial seeding, $10^4$ cells /well). On the day of analysis, cells were washed twice with DPBS, fixed in 4% PFA for 10 min at RT, and washed again twice with DPBS. Cells were then incubated in 70% ethanol at 4 °C for at least 1 hr and then washed with 1 ml of Wash Buffer A (LGC Biosearch Technologies) at room temperature for 5 min. Cells were subsequently hybridized with 100 µl of Hybridization Buffer (LGC Biosearch Technologies) containing the smRNA-FISH probes at a 1:100 dilution in a humid chamber at 37 °C o/n (not more than 16 h). The next day, cells were washed with 1 ml of Wash Buffer A at 37 °C for 30 min and stained with Wash Buffer A containing 10 µg/ml Hoechst 33342 at 37 °C for 30 min. Cells were then washed with 1 ml of Wash Buffer B (LGC Biosearch Technologies) at RT for 5 min, mounted with ProLong Gold (Thermo, P10144), and left to curate at 4 °C o/n before proceeding to image acquisition. Oligonucleotides probes were designed with the Stellaris smRNA-FISH probe designer (LGC Biosearch Technologies,

version 4.2), labeled with Quasar 570 and produced by LGC Biosearch Technologies. smRNA-FISH probes sequences are listed in *Supplementary file 3*.

## smRNA-FISH imaging

Image acquisition was performed using a DeltaVision Elite widefield microscope with an Olympus UPlanSApo 100 x /1.40-numerical aperture oil objective lens and a PCO Edge sCMOS camera. Z-stacks of 200 nm step size capturing the entire cell were acquired. Images were deconvolved with the built-in DeltaVision SoftWoRx Imaging software and maximum intensity projections were created. RNA-FISH foci were then quantified manually considering the overlap with Hoechst (nuclear fraction) and calculating *T-REX17* background staining (cytoplasmic fraction) using ImageJ (*Rueden et al., 2017*) and Fiji (*Schindelin et al., 2012*).

## Cell fractionation followed by RT-PCR and agarose gel band quantification

hiPSCs WT cells were differentiated to definitive endoderm cells (s. *Definitive endoderm (EN) differentiation*) and cytoplasmatic, nucleoplasmatic and chromatin fractions subsequently isolated utilizing the Subcellular Protein Fractionation Kit for Cultured Cells (Thermo Fisher Scientific, 78840) according to the manufactures protocol. Kit-provided buffers were substituted with 1 U/μl SUPERaseIn RNase Inhibitor (ThermoFisher Scientific, AM2694). RNA of respective cell fraction was isolated subsequently followed by cDNA synthesis (s. *RNA isolation and cDNA synthesis*). Relative PCR-product band intensity was obtained from agarose gel purified PCR-products utilizing the BioRad ChemiDoc XRS + imaging system. Band intensities of each fraction were normalized on the cytoplasmatic fraction (Cyt). Relative fracions per replicate were summed up to 100% before representing them as relative percentage fraction *Supplementary file 1*. PCR-primer sequences are listed in *Supplementary file 3*.

## Staining for FACS analysis

Undifferentiated or differentiated ZIP13K2 cultures were treated with Accutase for 15 min, 37 °C, 5% $CO_2$ to obtain a single-cell suspension. To quench the dissociation reaction and to wash the cells, FACS-buffer was added (Final DPBS, 5 mM EDTA (Thermo Fisher Scientific, 15575020), 10% Fetal bovine serum (FBS, PAN Biotech, P30-2602)). Next, cells were spun down at 300 x g, 5 min at 4 °C. Cells were then resuspended in FACS-buffer containing surface marker antibodies (s. *Supplementary file 3*) and incubated for 15 min at 4 °C in the dark. For extracellular stainings (ECS) only, cells were further washed once with FACS-buffer and spun down at 300 x g before FACS analysis was performed. If additional intracellular stainings (ECS +ICS) were performed, cells were washed once with FACS-buffer, supernatants were removed and cells fixed according to the manufacturer's instructions utilizing the True-Nuclear Transcription Factor Buffer Set (Biolegend, 424401). Intracellular staining was performed according to manufacturer's instructions before FACS analysis was carried out. ICS antibody dilutions are listed in *Supplementary file 3*. FACS analysis was performed on the FACS-Celesta Flow Cytometer (Beckton Dickinson). Raw data were analyzed using FlowJo (LLC) V10.6.2.

## Western blot and band quantification

Undifferentiated or differentiated ZIP13K2 cultures were treated with Accutase for 15 min, 37 °C, 5% $CO_2$ to obtain a single suspension. Single cell suspensions were washed once with ice cold DPBS and spun down at 300 x g, 5 min at 4 °C. Supernatants were removed and cell lysates generated by treatment for 30 min on ice with RIPA buffer (Thermo Fisher Scientific, 89900) supplemented with 1 x HALT protease inhibitor (Thermo Fisher Scientific, 87786). Lysates were spun down at 12,000 x g, 10 min at 4 °C and supernatants quantified for protein content using the Pierce BCA Protein Assay Kit (Thermo Fisher Scientific, 23227) according to the manufacturer's instructions.

For western blot, 20 μg total protein extract per sample were boiled in final 1 x Laemmli Buffer (BioRad, 1610747) containing 10% 2-Mercaptoethanol (M6250, Sigma-Aldrich) for 10 min at 95 °C, followed by cooling on ice for 5 min. Samples were then loaded on a NuPAGE 4–12%, Bis-Tris, 1.0 mm, Mini Protein Gel (Thermo Fisher Scientific, NP0322BOX) and ran at 200 V for 30 min in 1 x NuPAGE MOPS SDS Running Buffer (Thermo Fisher Scientific, NP0001) containing 1:400 NuPAGE Antioxidant (Thermo Fisher Scientific, NP0005). Protein transfer has been performed utilizing the iBlot 2 Starter

Kit, PVDF (Thermo Fisher Scientific, IB21002S) following the manufacturer's instructions for the P0 program.

PVDF membranes containing transferred proteins were incubated in blocking buffer (1 x TBS-T (Thermo Fisher Scientific, 28360), 5% Blotting-Grade Blocker (BioRad, 1706404)) for 1 hr at RT. Incubation with primary antibody dilution (s. *Supplementary file 3*) was performed in blocking buffer at 4 °C overnight. The following day, membranes were washed three times 10 min at RT with 1 x TBS-T and incubated for 2 hr at RT in secondary antibody dilution in blocking buffer (*Supplementary file 3*). Next, membranes were washed three times for 10 min at RT with 1 x TBS-T and developed using the SuperSignal West Dura Extended Duration Substrate (Thermo Fisher Scientific, 34075) according to the manufacturer's instructions and imaged on the BioRad ChemiDoc XRS+imaging system to finally obtain relative band-intensities. JNK or pJNK band-intensities were then normalized on their respective GAPDH levels before calculating relative pJNK levels (pJNK/JNK). Relative pJNK levels of EN time-course differentiations (*Figure 4C*, left panel; *Figure 4—figure supplement 2A*) were finally calculated and depicted as $Log_2FC(sgT-REX/sgCtrl)$ (*Figure 4C*, right panel). Raw data and calulcations are provided in *Supplementary file 1*.

## Computational analysis

Command-line processing of BAM, BED and bigwig files was done using SAMtools (v1.10) (*Li et al., 2009*), BEDtools (v2.25.0) (*Quinlan and Hall, 2010*) and UCSCtools (v4) (*Kuhn et al., 2013*). If not stated otherwise: All statistics and plots are generated using R version 3.6.0 and 3.6.1. In all boxplots, the centerline is median; boxes, first and third quartiles; whiskers, 1.5 x inter-quartile range; data beyond the end of the whiskers are displayed as points.

### Human vs. mouse *T-REX17* conservation analysis

Local alignment was performed with EMBOSS Water (*Madeira et al., 2022*). Visualizations were created with Matplotlib (*Hunter, 2007*). Alignment sequences were read into python using the Biopython library (*Cock et al., 2009*). The full sequence of the human *T-REX17* locus was aligned to the full sequence of the mouse *T-rex17* locus using Water. Aligned subsequences of 20 base pairs or more in length, including substitutions but excluding indels were used to calculate conservation. Additionally, individual exons and the enhancer sequence were also aligned with Water. Conserved stretches were connected from the human sequence box to the mouse sequence box and visualized as lines.

### 4Cseq data analysis

The raw sequencing reads were trimmed by using cutadapt (*Martin, 2011*) (--discard-untrimmed -e 0.05 m 25) to remove primer sequences and restriction enzyme sequences. The reads not matching those sequences, were removed from further analysis. The remaining reads were then mapped to the reference sequences GRCh37/hg19 by bowtie2 (*Langmead and Salzberg, 2012*) (default parameters). An iterative mapping procedure was performed. Specifically, the full-length reads were first mapped to the genome. The unmapped reads were then cut by 5-nt from the 3-prime end each time until they were successfully mapped to the genome or until they were shorter than 25 bp. The final mapped reads were assigned to valid fragments. The fragment counts were then normalized by RPM (reads per million) and smoothed by averaging the counts of the closest five fragments.

### Coding potential calculation

Whole genome multiple species alignments of 46 vertebrate species with human (assembly hg19, October 2009) as a reference have been retrieved from the UCSC genome browser (*Kent et al., 2002*). Human lincRNA annotation was obtained from Gencode (*Frankish et al., 2019*) (gencode.v33lift37.long_noncoding_RNAs.gtf, December 2019). All ORFs in each transcript were identified and the corresponding multiple species alignment was scored by the omega method of PhyloCSF (*Lin et al., 2011*; *Figure 2C*, left panel) shows 95% (2.5–97.5percentile) of the 271,572 sORFs from the (*Kent et al., 2002*) analyzed human lincRNAs (randomly sampled from chromosomes 16,21,18,11,17,5,10,19,22,2,7,X,12,6,Y). The *SOX17* CDS and all identified sORFs in *T-REX17* were scored by omega phyloCSF as shown in *Figure 2C*, right panel.

## RNA-seq

All RNAseq samples were pre-processed using cutadapt (*Martin, 2011*) to remove adapter and trim low quality bases. Reads were subsequently aligned against the human reference genome hg19 using STAR (*Dobin et al., 2013*) (parameter: outSAMtype BAM SortedByCoordinate `--outSAMattributes` Standard `--outSAMstrandField intronMotif --outSAMunmapped` Within `--quantMode` GeneCounts). Finally, Stringtie (*Pertea et al., 2015*) was used for calculation of strand-specific TPMs.

Differential gene expression was calculated using DESeq2 (*Love et al., 2014*). Genes with an absolute $\log_2$ fold change >1 and an adjusted *P*-value <0.05 were termed differentially expressed. Lowly expressed genes (all sample have a TPM <1) were excluded from the analysis.

## Capture Hi-C

Raw sequence reads of capture Hi-C (cHi-C) were mapped to the hg19 version of the human genome using BWA (v0.7.17-r1188) (*Li and Durbin, 2009*) with parameters (mem -A 1 -B 4 -E 50 L 0). Mapped reads were further processed by HiCExplorer (v3.6) (*Ramírez et al., 2018*) to remove duplicated reads and reads from dangling ends, self-circle, self-ligation and same fragments. The replicates were merged to construct contact matrices of 1 kb resolution. Normalization was performed to ensure that all samples have the same number of total contacts, followed by KR correction. The relative contact difference between two cHi-C maps was calculated by subtracting one from the other using the corrected matrices.

## SOX17 chromatin immunoprecipitation

The ChIP-seq sequencing data as well as the control input sequencing were aligned to the human reference genome (hg19) using BWA mem (*Heng, 2013*) using the default parameter. GATK (*McKenna et al., 2010*) was used to obtain alignment metrics and remove duplicates. Peaks were called using the MACS2 (2.1.2_dev) (*Zhang et al., 2008*) peakcall function using default parameters. After validation of replicate comparability and quality, replicates were merged on read level and reprocessed together with input samples. Background subtracted coverage files were obtained using MACS2 bdgcomp with -m FE. Peaks were removed from the analysis if overlapping with ENCODE blacklisted (hg19-blacklist.v2.bed) regions.

## GATA4/6 chromatin immunoprecipitation

The ChIP-seq sequencing data as well as the Fastqs for GATA4/6 ChIP-seq experiments were processed using the ENCODE ChIP-seq pipeline version 1.6.1 (https://github.com/ENCODE-DCC/chip-seq-pipeline2, copy archived at swh:1:rev:ec4295c8ac68be25b25357038d82ec942ac0bf8d; *Jin, 2022*) using default settings with the hg19 genome. Standard ENCODE ChIP-seq reference files were used as found in https://storage.googleapis.com/encode-pipeline-genome-data/genome_tsv/v1/hg19_caper.tsv. Pooled fold-change bigWigs were used.

## Single-cell RNAseq pipeline

Publicly available single-cell RNAseq raw data of already filtered 1195 cells from a gastrulating human embryo (*Tyser et al., 2021*) was downloaded from ArrayExpress (*Athar et al., 2019*) under accession code E-MTAB-9388. The GENCODE (*Frankish et al., 2021*) human transcriptome (GRCh37.p13) and its annotation were downloaded and added with the *T-REX17* entry. After building the transcriptome index, the transcripts abundance was quantified via Salmon v1.6.0 (*Patro et al., 2017*) in quasi-mapping-based mode using the –seqBias and the –gcBias flags. Data was loaded as a scanpy v1.4.4 (*Wolf et al., 2018*) object, reproducing clustering as reported by Tyser, R. C. v. et al. (*Tyser et al., 2021*). The resulting clusters were visualized via the scanpy UMAP representation in two dimensions, using default parameters (tl.umap). UMAPs are displayed in *Figure 2—figure supplement 1E* (upper panel).

## Bulk measurements from scRNAseq pipeline

To measure *T-REX17* read counts in endoderm cells fastq files were combined in one bulk raw file. The file went through a bulk RNAseq pipeline comprising a pre-alignment quality control via fastQC

v0.11.9, adaptor and low-quality bases trimming using cutadapt (*Martin, 2011*), post-QC and reads alignment against the human genome (GRCh37.p13) by means of STAR (*Dobin et al., 2013*) (parameters: `--outSAMtype BAM SortedByCoordinate, --chimSegmentMin 20, --outSAMstrandField intronMotif, --quantMode GeneCounts`). Finally, the BAM file was visualized using the Integrative Genomic Viewer (IGV) (*Robinson et al., 2011*). IGV tracks are displayed in *Figure 2—figure supplement 1E* (lower panel).

## Oxford Nanopore RNA analysis

All Oxford Nanopore Technologies derived runs were processed using the Nanopype pipeline (v1.1.0) (*Giesselmann et al., 2019*). The basecaller Guppy (v4.0.11) was used with the r9.4.1 high-accuracy configuration. Quality filtering was disabled for any base calling. Base-called reads were aligned against the human reference genome hg19 using minimap2 (v2.10) (*Li and Birol, 2018*) with the Oxford Nanopore Technologies parameter preset for spliced alignments (-ax splice -uf -k14). Only unique alignments (-F 2304) are reported.

## Oxford Nanopore RNA split-read analysis

Nanopore post processed split read data (*s. Oxford Nanopore RNA analysis*) from wild-type endoderm mRNA (*s. Extraction of polyA RNA for Nanopore sequencing; s. Preparation of Nanopore sequencing libraries*) were extracted from the junctions-track of BAM files visualized using the Integrative Genomic Viewer (IGV) (*Robinson et al., 2011*) utilizing the coordinates hg19, chr8:55115873–55141447. Split reads between hg19, chr8:55140806 (5'-sequence of Exon 1, *s. 5'/3' RACE PCR experiments*) and hg19, chr8: 55125601 (3'-sequence of Exon 3, *s. 5'/3' RACE PCR experiments*) were accounted for isoform Ex1 +2 (s. *Figure 1—figure supplement 1C*). Full isoform Ex1 +2 sequence (~2,8 kb long) can be found in *Supplementary file 1*.

Split reads between hg19, chr8:55140806 (5'-sequence of Exon 1, *s. 5'/3' RACE PCR experiments*) and hg19, chr8:55123254 (3'-sequence of Exon 3, *s. 5'/3' RACE PCR experiments*) were accounted for isoform Ex1 +3 (s. *Figure 1—figure supplement 1C*). All other reads were accounted as "sloppy spliced" reads and together with both isoforms calculated in relative terms (s. *Figure 1—figure supplement 1C*). Full isoform Ex1 +3 sequence (~3 kb long) can be found in *Supplementary file 1*. Summary of the relative isoform quantification is displayed in *Figure 2F*.

## Mass spectrometry analysis and ranking of *T-REX17* protein partners

Raw MS data were processed with MaxQuant software (v 1.6.10.43) and searched against the human proteome database UniProtKB with 75,074 entries, released in May 2020. Parameters of MaxQuant database searching were a false discovery rate (FDR) of 0.01 for proteins and peptides, cysteine carbamidomethylation was set as fixed modification, while N-terminal acetylation and methionine oxidation were set as variable modifications. Protein abundance in each of the three samples has been quantified by calculating Label free quantitation (LFQ) values for each detected protein. Protein targets have then been ranked based on the $Log_2[(Even_{LFQ} + Odd_{LFQ})/2/LacZ_{LFQ}]$ extrapolated values.

## Plotting

Plots were generated with GraphPad Prism 8, R 3.6.0 and R 3.6.1.

## Acknowledgements

We are grateful for the support and feedback received by all Meissner Lab members during the project development. Special recognition to T Aktas and IA Ilik for experimental advising. AA Hernandez for the *T-REX17* naming. B Lukaszewska-McGreal for support with the mass spectrometry experiment. RD Acemel for suggestions regarding the virtual 4 C analysis. I Ulitsky for advice with the conservation analysis. B Fauler for help with microscopy; D Ibrahim for fruitful feedback; MPIMG Seq-Core for NGS support. This work was funded by the NIH (DP3K111898 R.M and A.M; P01GM099117 J.R. and A.M.) and the Max Planck Society.

## Additional information

### Funding

| Funder | Grant reference number | Author |
| --- | --- | --- |
| National Institutes of Health | P01GM099117 | John L Rinn<br>Alexander Meissner |
| National Institutes of Health | DP3K111898 | René Maehr<br>Alexander Meissner |
| Max Planck Society | | Alexander Meissner |

The funders had no role in study design, data collection and interpretation, or the decision to submit the work for publication. Open access funding provided by Max Planck Society.

### Author contributions

Alexandro Landshammer, Conceptualization, Data curation, Formal analysis, Validation, Investigation, Visualization, Methodology, Writing – original draft, Writing – review and editing; Adriano Bolondi, Conceptualization, Data curation, Formal analysis, Supervision, Validation, Investigation, Visualization, Methodology, Writing – original draft, Writing – review and editing; Helene Kretzmer, Resources, Data curation, Formal analysis, Visualization, Methodology; Christian Much, Alina Rose, Bjoern Braendl, Krishna Mohan Parsi, Investigation; René Buschow, Hua-Jun Wu, Sebastian D Mackowiak, Pay Giesselmann, Rosaria Tornisiello, Jack Huey, Formal analysis; Thorsten Mielke, René Maehr, Denes Hnisz, Franziska Michor, John L Rinn, Supervision; David Meierhofer, Formal analysis, Investigation, Methodology; Alexander Meissner, Conceptualization, Supervision, Funding acquisition, Writing – review and editing

### Author ORCIDs

Alexandro Landshammer http://orcid.org/0000-0001-5367-3303
Adriano Bolondi http://orcid.org/0000-0002-1096-9435
René Buschow http://orcid.org/0000-0002-9800-2578
Sebastian D Mackowiak http://orcid.org/0000-0003-1673-5389
Krishna Mohan Parsi http://orcid.org/0000-0002-6002-3816
René Maehr http://orcid.org/0000-0002-9520-3382
John L Rinn http://orcid.org/0000-0002-7231-7539
Alexander Meissner http://orcid.org/0000-0001-8646-7469

### Decision letter and Author response

Decision letter https://doi.org/10.7554/eLife.83077.sa1
Author response https://doi.org/10.7554/eLife.83077.sa2

## Additional files

### Supplementary files

• Supplementary file 1. List of external data sets used in this study. Luciferase assay raw and normalized data for *eSOX17*, *eSOX17.1*, *eSOX17.1*, *pSOX17* and *pT-REX17* of respective day of endodermal differentiation. Cell fractionation RT-PCR relative band-intensity quantification. MinION split reads extracted from IGV junctions-track BAM files. *T-REX17* isoform sequences. Wetsern Blot relative band-intensity quantification.

• Supplementary file 2. RNAseq data set with TPM values from undifferentiated (iPSC), day 3 and day 5 endoderm differentiated sgCtrl and sgT-REX17 CRISPRi cells. Differential gene expression analysis of RNAseq data between sgCtrl and sgT-REX17 from respective days of endoderm differentiated cells. RNAseq data sets with TPM values from day 9 pancreatic precursor (PP) differentiated sgCtrl and sgT-REX17 CRISPRi cells. Differential gene expression analysis of RNAseq data between sgCtrl and sgT-REX17 from respective days of pancreatic precursor differentiation.

• Supplementary file 3. Taqman qRT-PCR probes used in this study. qRT-PCR primers, sgRNA oligonucleotides, cloning and genotyping primers used in this study. Antibodies and their respective application specific dilutions used in this study. *T-REX17* exonic smFISH probes and their sequence

used in this study. *SOX17* locus capture HiC probes and their sequence used in this study. RNA-pulldown probes and their sequences used in this study.

• Supplementary file 4. H3K9me3 ChIP-qPCR raw and normalized data from sgCtrl and sgT-REX17 day 5 endoderm cells. Raw data of $Log_{10}$ (ECAD/NCAD) signal ratios and respective quantification from day 5 sgCtrl and sgT-REX17 endoderm cells. Raw data of VIMENTIN signal and respective quantification from sgCtrl and sgT-REX17 undifferentiated iPSCs and day 5 endoderm cells. Raw data of PDX1 signal and respective quantification from day 9 sgCtrl and sgT-REX17 pancreatic progenitor cells. List of putative *T-REX17* interaction partners as measured in the RNA-pulldown followed by mass spectrometry.

• MDAR checklist

• Source code 1. Source code for the 4C analysis in *Figure 1.*

• Source code 2. Source code for the conservation analysis in *Figure 1—figure supplement 1*.

### Data availability

All data presented in this study are available in the main text, methods or tables. Sequencing data have been deposited in the Gene Expression Omnibus (GEO) under accession code GSE178990. Source data files have been provided where necessary. Codes used to perform the analysis in this study are available at the following GitHub locations or as source data files in the submission documents: https://github.com/RosariaTornisiello/lncSox17.git (copy archived at swh:1:rev:3cfa5a270e-27c291d0dd390173e2311406c78bd7); https://github.com/Drmirdeep/stitch_maf (copy archived at swh:1:rev:706a7b94e0be9b02a10f47680b9a693f1178e409); https://github.com/Drmirdeep/micpdp (copy archived at swh:1:rev:901d60c3b78208bef072ff0906e6ec271f1f9667); https://github.com/HeleneKretzmer/lncRNA_Sox17 (copy archived at swh:1:rev:6d5d930e306ec8eb-b4420609aa30986429e474be); https://github.com/ENCODE-DCC/chip-seq-pipeline2 (copy archived at swh:1:rev:ec4295c8ac68be25b25357038d82ec942ac0bf8d).

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
