## [Editor Report]

Supported by a large set of complementary experiments, the authors convincingly show that the lncRNA *T-REX17* is required for human definitive endoderm differentiation. *T-REX17* function is not related to the adjacent *SOX17* gene that lies in the same topological domain (TAD), implying a trans-acting role. The study is important because it sheds light on the stage-specific role of lncRNAs in cell lineage induction.

---

## [Decision Letter]

**Decision letter after peer review:**

Thank you for submitting your article "Discovery and characterization of *LNCSOX17* as an essential regulator in human endoderm formation" for consideration by *eLife*. Your article has been reviewed by 3 peer reviewers, and the evaluation has been overseen by a Reviewing Editor and Marianne Bronner as the Senior Editor. The following individuals involved in review of your submission have agreed to reveal their identity: Pablo Navarro (Reviewer #2); Peter J Rugg-Gunn (Reviewer #3).

Comments from the three reviewers were overall positive and broadly similar. The large breadth of relevant approaches that the authors used to tackle the function of this lncRNA were greatly appreciated, even though the mechanism by which it controls endoderm differentiation stays unclear. From their individual evaluation and the common discussion that followed, it was agreed that the requested revisions should be minor-statistical analyses, and text adjustments and clarifications-with the idea of strengthening and clarifying some of the findings of the manuscript.

More specifically, the essential revisions are:

1. Provide the exact genomic coordinates of the lncRNA.

2. Perform statistical analyses (DEG analyses in particular).

3. Revise the naming of LNCSOX17: it is misleading as there is no functional connection with SOX17.

4. Use the reviewers ‘ comments to clarify the text, including pieces of additional information that are important to include in the text.

Please answer the reviewers ' comments in a point-by-point rebuttal letter. Additional experiments are not necessarily expected, but the reviewers felt that if one had to be done, it would be the test of the trans-acting function of LNCSOX17 through ectopic expression (With the appreciation that the length and variable splicing patterns of the lncRNA might make this difficult).

*Reviewer #1 (Recommendations for the authors):*

The points to be addressed before proceeding to final publication are listed below and divided in Major and Minor corrections. We hope the authors find them useful.

Corrections

Page 5: If lncRNA has 40 foci/cell it is unlikely that is a cis-acting lncRNA. Indeed, does RNA-seq after depletion of LNCSOX17 leads to any changes in genes associated within the same TAD (beside SOX17 not changing)?

Page 7: it was not clear whether LncSox17 P(A)/P(A)cells (Supp Figure 3I-K) that also loss LNCSOX17 expression exhibit the same differentiation phenotype as cells depleted of lncRNA using the dCAS9-KRAB-MeCP2 method? Could the authors confirm endoderm defects with IF (eg ECAD, NCAD, VIM) with this CRISPR PAS system? Do they also have *JNK* signaling pathway activated in LncSox17 PAS cells? What about pancreatic progenitor marker genes?

At the moment the endoderm defects are confirmed only with single guide RNA using the dCAS9-KRAB-MeCP2 system to deplete LNCSOX17 as shown in Figure 4.

Page 6: why not targeting LNCSOX17 with LNA gapmers against different exons since it is a nuclear lncRNA (targeting the 5' and 3') ? Although the authors mention that the CAS9-KRAB-MeCPD did not affect SOX17 (Supp 3c), did the authors check the expression of other neighboring genes upon dCAS9-KRAB-MeCP2? In addition, at least 2-3 guide RNAs should be used even with CRISPR to avoid any off target effects and to assure phenotype is reproducible. Do stable cell lines expressing dCAS9-KRAB-MeCP2 differentiate normally before adding control guides and guide against LNCSOX17? More importantly, does depletion of SOX17 affects endoderm differentiation? This should be stated clearly (Supp 3H).

Page 48, Figure 2C – The authors should consider adding RNA fractionation experiments to the panel to assess whether LNCSOX17 is associated to chromatin. The result will also give indications on the possible lncRNA molecular mechanisms (i.e. nucleoplasm vs chromatin enrichment).

Page 63, Supp Figure 4A – The prediction of the RNA secondary structure is superficial and it does not support the data presented later on. Moreover, although I understand that Vienna RNA has a maximum limit of nucleotides to input, I find useless that the prediction is run only on the spliced isoforms, representing only the 20% of the LNCSOX17 transcripts (the authors mention at Page 6 that the gene undergoes sloppy splicing). I suggest removing this part.

Page 8 and Page 63, Supp Figure 4D – It is not clear why the authors focus only on the hnRNP family, given the high number of putative candidates upon mass spectrometry analysis. Please clarify this point. A gene ontology analysis could help data interpretation.

Moreover, it is not clear whether the association with hnRNP proteins is somehow functional to guarantee endoderm differentiation. Thus, the phenotype of LNCSOX17 interacting proteins such as hnRNPU should be to further investigated to reinforce the role of LNCSOX17 – hnRNPU in the context of endoderm differentiation. This is important since lncRNA- bound proteins identified in this manuscript are found to interact with many other lncRNAs. Also interaction between hnRNPU and LNCSOX17 should be confirmed with CLIP, not RIP. RIP qPCR is not ideal method for RNA-protein interaction since it is known to lead to post-lysis reassociation of RBP and RNAs. Please see PMID: 15388877.

Page 52, Figure 4B – RNA-seq in LNCSOX17-depleted cells at the 5 day of endoderm differentiation reveals a high number of differentially expressed genes, however Figure 2A shows that the lncRNA is expressed already after 3 days from the start of endoderm differentiation. The authors should comment this in the discussion (e.g. how the dynamics of such events are related to each other, if the case?).

Page 9: How did the authors decide to focus on CXCR4 since it was introduced before RNA-seq analysis?

*Reviewer #2 (Recommendations for the authors):*

I wonder if LncSox17 could not be TCONS_00014700 (hg38:chr8:54379158-54380837; lincRNA and TUCP transcripts from UCSC) and have any functional relationship with ENSG00000286471 (GencodeV41).

Perhaps a simultaneous smFISH using both intronic and exonic probes for both Sox17 and LncSox17 would clarify this issue and perhaps reveal more subtle regulations between the two genes.

*Reviewer #3 (Recommendations for the authors):*

1. Figure 1A: It looks like the LNCSOX17 locus has fairly high levels of DNA methylation in PSC and then low methylation as the gene is activated in EN. This contrasts with SOX17 that is unmethylated in both cell types. LNCSOX17 can therefore perhaps provide an interesting and relatively atypical example of how a developmental gene is presumably not restricted by Polycomb-mediated processes but rather by a specific DNA demethylation event in the forming endoderm. Can the authors comment on this?

2. Figure Supplement 1A: The sequence conversation between human and mouse LNCSOX17 appears moderate in exon 1 and the enhancer region, but very low in the rest of the transcript. Also, the mouse RNA-seq track is very noisy, perhaps due to the scaling applied. From the figure provided, I found it difficult to tell to whether the LNCSOX17 really was present and meaningfully expressed in these other species, and if it is then what is the level of conservation of the transcripts and secondary structures between species?

3. I could not see a quantitative assessment of LNCSOX17 expression compared to SOX17 in human EN cells. Can the authors please show log2 RPKM values (or similar)?

4. Figure 3D: It is interesting that SOX17 still seems to be binding to the eSOX17 even in the CRISPRi cells when presumably the region is in a heterochromatinised state (Figure Supplement 3C). One prior possibility was that the transcription of LNCSOX17 was needed to keep the region accessible, engage in long distance interactions, and facilitate SOX17 occupancy. But this experiment suggests that is probably not the case. I felt this point could be made in the text.

5. Figure Supplement 3H: The deletion of SOX17 lead to the failure to induce LNCSOX17, but it is unclear if that was due to failure of SOX17 to directly activate LNCSOX17 (as currently implied), or due to a general failure to make EN and therefore an indirect effect on LNCSOX17. Do the SOX17 knockout cells make EN cells?

6. The proposed model, that LNCSOX17 is not acting in cis and therefore potentially in trans, is well supported by several experiments. But one further strong piece of evidence for a trans-acting function would be if the LNCSOX17 mutant phenotype could be rescued by expressing LNCSOX17 ectopically. Have the authors tried this? With the appreciation that the long transcript and variable splicing might make this difficult.

7. I could not really follow why the transcript was called LNCSOX17. Presumably because it lies within the same TAD as SOX17? Nevertheless, I feel this name could be confusing because, as the authors show, LNCSOX17 does not have much to do with SOX17. The authors might want to consider proposing an alternative name for their transcript to avoid confusion over inferring a genetic or functional link to SOX17.

---

## [Author Response]

Essential revisions:1. Provide the exact genomic coordinates of the lncRNA.

We now provide this information in the legend of Figure 1.

2. Perform statistical analyses (DEG analyses in particular).

We now provide this information in the figure legend and methods section.

3. Revise the naming of LNCSOX17: it is misleading as there is no functional connection with SOX17.

We revised the name *LNCSOX17* to *T-REX17* (Transcript Regulating Endoderm and activated by soX17) and adapted the new naming in all text, figures and tables. Please note that we changed the name also in the point by point response to reviewer comments. Moreover, we had to adapt our title to include the “non-coding RNA” since the new name does not contain this information anymore.

4. Use the reviewers ‘ comments to clarify the text, including pieces of additional information that are important to include in the text.

We have done this throughout. Please, see below our point-by-point response to all reviewers’ comments.

Please answer the reviewers ' comments in a point-by-point rebuttal letter. Additional experiments are not necessarily expected, but the reviewers felt that if one had to be done, it would be the test of the trans-acting function of LNCSOX17 through ectopic expression (With the appreciation that the length and variable splicing patterns of the lncRNA might make this difficult).

We agree with this assessment and have tried this particular experiment, but run into the largely expected difficulties. Please see below response to reviewer 3 (point 6, page 26, lines 666-703 of this document).

Reviewer #1 (Recommendations for the authors):The points to be addressed before proceeding to final publication are listed below and divided in Major and Minor corrections. We hope the authors find them useful.CorrectionsPage 5: If lncRNA has 40 foci/cell it is unlikely that is a cis-acting lncRNA. Indeed, does RNA-seq after depletion of LNCSOX17 leads to any changes in genes associated within the same TAD (beside SOX17 not changing)?

As shown in Figure 2D and correctly pointed out by the reviewer we observe a median amount of 40 foci/cell for *T-REX17* in endoderm cells:

This type of foci distribution is unlikely to reflect a *cis-*acting function, as also acknowledged by the reviewer and noted in our discussion.

*SOX17* is currently the only annotated gene present in the loop-domain as shown in our Figure 1A (more details on the TAD structure, its genetic and epigenetic landscape can also be found in one of our previous publications: PMID: 34385432):

Furthermore, our data show that *T-REX17* (or its absence) does not influence *SOX17* expression levels (see Figure 3 and Figure Supplement 3).

Taken together, all evidence to date argues against a *cis*-function within the domain.

In the revised version, we now expanded our analysis to also include genes present in the neighboring TAD and show that their expression in the absence of *T-REX17* is largely unaffected:

We included a panel in Figure Supplement 3E and adjusted the text (page 7 lines 192-194) to describe this analysis:

“We also confirmed that unrelated genes present in neighboring domains were unaffected by the perturbation (Figure supplement 3E).”

Page 7: it was not clear whether LncSox17 P(A)/P(A)cells (Supp Figure 3I-K) that also loss LNCSOX17 expression exhibit the same differentiation phenotype as cells depleted of lncRNA using the dCAS9-KRAB-MeCP2 method? Could the authors confirm endoderm defects with IF (eg ECAD, NCAD, VIM) with this CRISPR PAS system? Do they also have JNK signaling pathway activated in LncSox17 PAS cells? What about pancreatic progenitor marker genes?At the moment the endoderm defects are confirmed only with single guide RNA using the dCAS9-KRAB-MeCP2 system to deplete LNCSOX17 as shown in Figure 4.

We acknowledge that in our manuscript we characterized the cellular phenotype of *T-REX17* depleted cells only in the dCas9-KRAB-MeCP2 cell line at greater depth (Figure 4). Nevertheless, the p(A)/p(A) background line (which represses *T-REX17* without inhibiting locus transcription) displays a similar phenotypic outcome, which strongly supports the idea that the observed phenotype in the dCas9 line is not due to off-target effect of dCas9-KRAB-MeCP2 but rather hints at a functional role for *T-REX17*.

Briefly, we observed reduced levels of CXCR4 concomitant with preserved expression of SOX17 (Figure Supplement 5A), reduced expression of a set of endoderm specific genes (Figure Supplement 5G) and upregulation of a pluripotency-associated factor like NANOG (Figure Supplement 5G).

Importantly, all these phenotypes and specifically the endoderm specific DEGs, are also observed in the dCas9-KRAB-MeCP2 line (Figure Supplement 5H), reinforcing the interpretation of a convergent phenotypic alteration caused by *T-REX17* loss:

In the text, we refer to the phenotypes observed in both the dCas9-KRAB-MeCP2 repression and the p(A)/p(A) (pages 9-10, lines 249-252 and 261-267).

Page 6: why not targeting LNCSOX17 with LNA gapmers against different exons since it is a nuclear lncRNA (targeting the 5' and 3') ? Although the authors mention that the CAS9-KRAB-MeCPD did not affect SOX17 (Supp 3c), did the authors check the expression of other neighboring genes upon dCAS9-KRAB-MeCP2? In addition, at least 2-3 guide RNAs should be used even with CRISPR to avoid any off target effects and to assure phenotype is reproducible. Do stable cell lines expressing dCAS9-KRAB-MeCP2 differentiate normally before adding control guides and guide against LNCSOX17? More importantly, does depletion of SOX17 affects endoderm differentiation? This should be stated clearly (Supp 3H).

The reviewer is correct that interference with LNA or siRNA/shRNA remains a frequently used approach to study gene function in mammalian cells. However, nuclear transcripts—such as many long non-coding RNAs (lncRNAs)—may be more difficult to target in this way (PMID: 22955988, PMID: 24296535). Moreover, such an approach would require a transient transfection of differentiating endoderm cells, which we have had little success to date. As a result, we decided to focus on alternative methods such as dCas9-KRAB-MeCP2 based repression and early transcriptional termination cassette.

Page 48, Figure 2C – The authors should consider adding RNA fractionation experiments to the panel to assess whether LNCSOX17 is associated to chromatin. The result will also give indications on the possible lncRNA molecular mechanisms (i.e. nucleoplasm vs chromatin enrichment).

As shown above, in the revised version, we now expanded our analysis to also include genes present in the neighboring TAD and show that their expression in the absence of *T-REX17* is largely unaffected:

We included this panel in Figure Supplement 3E and adjusted the text (page 7 lines 192-194) to describe this analysis:

“We also confirmed that unrelated genes present in neighboring domains were unaffected by the perturbation (Figure supplement 3E).”

Page 63, Supp Figure 4A – The prediction of the RNA secondary structure is superficial and it does not support the data presented later on. Moreover, although I understand that Vienna RNA has a maximum limit of nucleotides to input, I find useless that the prediction is run only on the spliced isoforms, representing only the 20% of the LNCSOX17 transcripts (the authors mention at Page 6 that the gene undergoes sloppy splicing). I suggest removing this part.

We agree and removed this analysis from the figure, and adjusted the main text, methods and figure legend accordingly (pages 8, 44 and 66, lines 222-226, 1109-1113 and 1432-1433).

Page 8 and Page 63, Supp Figure 4D – It is not clear why the authors focus only on the hnRNP family, given the high number of putative candidates upon mass spectrometry analysis. Please clarify this point. A gene ontology analysis could help data interpretation.

The reviewer is correct and a better rationale should be provided. We also tried performing the GO analysis using the enriched candidate list, but could not find any significant terms with obvious links to endoderm development or developmental progression. Most terms point to rather general cellular functions (Author response image 1):

**Author response image 1. sa2fig1:** Gene ontology (GO) analysis of the candidate T-REX17 protein interactors as measured by RNA pulldown followed by mass spectrometry. Blue bars represent significant (FDR<0.05) biological process identified. The analysis was performed using WEB-based GEne SeT AnaLysis Toolkit (WebGESTALT).

As for the HNRNPs, these proteins have been well characterized as RNA interacting factors, often associating in multiprotein complexes involved in a variety of cellular processes (PMID: 24463464; PMID: 28636939; PMID: 20833368; PMID: 22325991; PMID: 22574288; PMID: 31350345; PMID: 31892844; PMID: 25406515). We therefore decided to independently validate the MS results with our RIP experiment (Figure Supplement 4E,F)

We now provide the full list of genes and their ranked enrichment in Table 4 to help manual inspection and further investigation of the complete list.

Moreover, it is not clear whether the association with hnRNP proteins is somehow functional to guarantee endoderm differentiation. Thus, the phenotype of LNCSOX17 interacting proteins such as hnRNPU should be to further investigated to reinforce the role of LNCSOX17 – hnRNPU in the context of endoderm differentiation. This is important since lncRNA- bound proteins identified in this manuscript are found to interact with many other lncRNAs.

We agree with the reviewer that we can’t establish a functional link between T-REX17-HNRNPU interaction and the observed phenotypes with the current available information. Therefore, we decided to rephrase the paragraph about the HNRNPU interaction to reflect this aspect (page 8, lines 221-242).

Also interaction between hnRNPU and LNCSOX17 should be confirmed with CLIP, not RIP. RIP qPCR is not ideal method for RNA-protein interaction since it is known to lead to post-lysis reassociation of RBP and RNAs. Please see PMID: 15388877.

The reviewer is correct that CLIP may be preferred in this context, though the RIP result shown in Figure Supplement 4F is convincing to us.

Page 52, Figure 4B – RNA-seq in LNCSOX17-depleted cells at the 5 day of endoderm differentiation reveals a high number of differentially expressed genes, however Figure 2A shows that the lncRNA is expressed already after 3 days from the start of endoderm differentiation. The authors should comment this in the discussion (e.g. how the dynamics of such events are related to each other, if the case?).

As correctly stated by the reviewer, the onset of *T-REX17* expression is first detectable by day 3 of endoderm differentiation in WT cells (Figure 2A; Figure 3B; Figure Supplement 5F).

Page 9: How did the authors decide to focus on CXCR4 since it was introduced before RNA-seq analysis?

We made this choice based on the literature (PMID: 16258519; PMID: 25693565; PMID: 34385432; PMID: 25843708; PMID: 27705785; PMID: 21358635; PMID: 24412311). Briefly, C-X-C chemokine receptor type 4 (CXCR4) is a chemokine receptor expressed by various cell types during development and homeostasis. During the early stages of human differentiation and exit from pluripotency, its expression is specifically confined in definitive endoderm, and for this reason this receptor has been widely adopted by the community as a marker for successful differentiation into the endoderm lineage.

Reviewer #2 (Recommendations for the authors):I wonder if LncSox17 could not be TCONS_00014700 (hg38:chr8:54379158-54380837; lincRNA and TUCP transcripts from UCSC) and have any functional relationship with ENSG00000286471 (GencodeV41).

TCONS_00014700 on chr8:54379158-54380837 is listed in the lincRNA and TUCP transcripts dataset (PMID: 21890647) cataloged by the Rinn Lab. It is a 1.6kb long transcript present within the SOX17 loop domain, transcribed on the (+)-strand and characterized by the presence of 2 exons. It is localized in proximity of the ENSG00000286471 transcript, but far away from the *T-REX17* genomic location (hg19, chr8:55117776-55140806). The difference in size (22kb vs 1.6kb), synthesis strand ((-) vs (+)), isoform configurations and genomic location (see above for *T-REX17* exact genomic location) suggest that these are two independent RNA molecules:

John Rinn (co-author on this manuscript) also had a closer look at this relationship and confirmed that the two transcripts do not occupy the same location. Moreover, neither ENSG00000286471 nor TCONS_00014700 are expressed in definitive endoderm where *T-REX17* is active (Figure 1A, we do not see any transcription at the locus except for *T-REX17* and SOX17), therefore we are confident that they are functionally independent elements.

Perhaps a simultaneous smFISH using both intronic and exonic probes for both Sox17 and LncSox17 would clarify this issue and perhaps reveal more subtle regulations between the two genes.

The experiments proposed by the reviewer could be helpful in clarifying a possible *cis-*regulative function for *T-REX17*. In Figure 1D we have shown that the two genomic locations interact in endoderm cells, since an active SOX17 enhancer is present in close proximity to the *T-REX17* promoter.

Based on this analysis, we would expect that intronic probes smFISH would highlight co-localization of the two nascent transcripts. Our current smFISH already provide some hints for the localization of the nascent and mature *T-REX17* transcripts, since the site of transcription shows an enriched signal as compared to the other foci (indicated by red arrows and detailed in the Figure 2C legend):

Exon probes would in contrast have no effect on the smFISH of *T-REX17* and would localize SOX17 prevalently in the cytoplasm (if using probes spanning exon-exon junctions). The foci distribution pattern of *T-REX17* would strongly indicate an activity within the nuclear compartment but at different genomic locations far from its transcriptional start site (also supported by our perturbation experiments in Figure 3 showing intact chromatin, expression and SOX17 occupancy in *T-REX17* depleted cells).

Reviewer #3 (Recommendations for the authors):1. Figure 1A: It looks like the LNCSOX17 locus has fairly high levels of DNA methylation in PSC and then low methylation as the gene is activated in EN. This contrasts with SOX17 that is unmethylated in both cell types. LNCSOX17 can therefore perhaps provide an interesting and relatively atypical example of how a developmental gene is presumably not restricted by Polycomb-mediated processes but rather by a specific DNA demethylation event in the forming endoderm. Can the authors comment on this?

The reviewer raises a very interesting point regarding the epigenetic control of developmental genes within the *SOX17* loop domain. *SOX17,* as most developmental regulators, contains CpG islands (CGIs) within its promoter which is always unmethylated irrespective of its expression status. As the reviewer suggests, its expression is rather controlled by Polycomb.

As for the *T-REX17* locus, it is worth highlighting that the large and tissue specific differentially methylated region (DMR) mainly overlaps with the *SOX17* active enhancer (*eSOX17*). Only few CpGs at the tail of the DMR overlap with the *T-REX17* promoter region.

Some developmental *cis-*regulatory elements are known to be regulated by DNA methylation, and loss of this modification is correlated with transcription factors binding and chromatin opening (PMID: 31422875, PMID: 25693565).

2. Figure Supplement 1A: The sequence conversation between human and mouse LNCSOX17 appears moderate in exon 1 and the enhancer region, but very low in the rest of the transcript.

The presence of orthologous transcripts in other species provides initial support for a functional role of lncRNAs. The reviewer is right when saying that most of the sequence conservation is localized in the first portion of the transcript. As lncRNAs often exert their function through their secondary structure it may not be entirely unexpected to see limited sequence conservation (PMID: 35098341). Other parameters such us synteny and positional conservation of neighboring genes (see T-REX17 and its position as compared to SOX17 in the analyzed clades) are emerging as more appropriate analysis for orthologous non-coding transcripts identification across species (PMID: 33563213; PMID: 31450588; PMID: 31247106)

Also, the mouse RNA-seq track is very noisy, perhaps due to the scaling applied. From the figure provided, I found it difficult to tell to whether the LNCSOX17 really was present and meaningfully expressed in these other species, and if it is then what is the level of conservation of the transcripts and secondary structures between species?

We agree with the reviewer that it is not trivial to establish a direct parallel between the transcripts in various species. We are of course limited here by the scarcity of high-quality in vivo definitive endoderm RNA-seq datasets from other species, including mouse.

As the reviewer is also hinting at in the last part of the comment, the secondary structure of the RNA might be more relevant than the actual sequence to infer functional conservation. Following suggestions from the first reviewer (see response above, lines 219-225 of this document) and given the unreliable secondary structure predicted for *T-REX17*, we now decided to remove the secondary structure analysis from our manuscript and therefore won’t be able to compare it to the mouse counterpart.

To highlight the limitation of this analysis, we rephrased the main text (page 4, lines 106-109) to reflect the reviewer comment:

Although the sequence conservation to the mouse is only modest, we detect the presence of a distal SOX17 transcript in a number of vertebrates based on stage- and tissue-matched embryonic data (Figure supplement 1A, left).”

3. I could not see a quantitative assessment of LNCSOX17 expression compared to SOX17 in human EN cells. Can the authors please show log2 RPKM values (or similar)?

In the current version of the manuscript, we use different approaches to measure the expression of SOX17 and *T-REX17* during endoderm formation. With qPCR (carried out using Taqman probes and therefore being quantitatively comparable across targets) we show the levels of the two transcripts with 24h resolution during in vitro differentiation (Figure 2A)

Moreover, we use a dataset from the NIH Roadmap Epigenomics Mapping Consortium (PMID: 20944595) to calculate the log_2_TPM for the two transcripts in stem cell-derived human endoderm cells and show this result in Figure 2B:

We now re-analyzed our internal day 5 endoderm RNA-seq dataset and show in Author response image 3 the log_2_TPM values for both *SOX17* and *T-REX17*.

**Author response image 3. sa2fig3:** 

Given that this result is in line with the previous quantitative analysis of *SOX17* and *T-REX17* gene expression (Figure 2A; Figure 2B; Figure Supplement 3J), we include this panel for the reviewers but do not include it in the revised manuscript. Moreover, TPM values for all our time-course RNA-seq analysis can be found in Supplementary Table 2.

4. Figure 3D: It is interesting that SOX17 still seems to be binding to the eSOX17 even in the CRISPRi cells when presumably the region is in a heterochromatinised state (Figure Supplement 3C). One prior possibility was that the transcription of LNCSOX17 was needed to keep the region accessible, engage in long distance interactions, and facilitate SOX17 occupancy. But this experiment suggests that is probably not the case. I felt this point could be made in the text.

This observation puzzled us as well for quite long time, and motivated us to introduce the pA/pA construct to preserve local transcription while depleting *T-REX17*. Blocking *T-REX17* transcription (dCas9-KRAB-MeCP2 repression) or the generation of its transcript (pA/pA) do not affect SOX17 expression. This indicates that indeed *eSOX17* can be activated (and theoretically bound by SOX17) independently of *T-REX17* function. Pioneer factor activity of SOX17 or other locally bound factors (such as FOXA2) could be a plausible explanation for its ability to bind *eSOX17* despite the induced heterochromatinization. We now updated the main text to specify this interesting point (page 7, lines 195-199)

“Next, we performed SOX17 Chromatin Immunoprecipitation sequencing (ChIP-seq) and show that SOX17 occupancy at the SOX17 locus (including at its induced heterochromatic distal enhancer (eSOX17)) as well as genome-wide is largely unaffected by the loss of T-REX17 (Figure 3D,F, Figure supplement 3F).”

5. Figure Supplement 3H: The deletion of SOX17 lead to the failure to induce LNCSOX17, but it is unclear if that was due to failure of SOX17 to directly activate LNCSOX17 (as currently implied), or due to a general failure to make EN and therefore an indirect effect on LNCSOX17.

The reviewer raises a relevant point here. Given the failure of SOX17-KO cells to properly differentiate into endoderm (see response at lines 181-203 of this document), it is challenging to distinguish between the two proposed scenarios for *T-REX17* activation (direct activation by SOX17 vs general failure in differentiation). Nonetheless, we note:

1) *T-REX17* starts to be expressed after SOX17 (Figure 2A)

2) SOX17 binds *T-REX17* promoter (Figure 3G)

3) SOX17-KO cells fail to activate *T-REX17* (Figure Supplement 3J)

In the revised manuscript we show that SOX17-KO cells fail to properly differentiate into endoderm and explicitly highlight it (page 7, lines 202-207).

“Notably, homozygous knockout cells fail to induce GATA4 expression and show no activation of T-REX17 (Figure supplement 3J).”

Do the SOX17 knockout cells make EN cells?

Depletion of SOX17 clearly impacts endoderm differentiation.

In the revised manuscript we now show that a SOX17 homozygous KO cell line fails to induce *GATA4* (and other marker genes) expression upon endoderm differentiation (Figure Supplement 3J):

Moreover, in the current manuscript we show that SOX17 enhancer disruption results in delayed SOX17 expression during endoderm differentiation and reduced CXCR4 levels (Figure Supplement 1F):

Based on published literature, SOX17 miss-expression causes major differentiation defects, as we (PMID: 34385432) and others have shown (PMID: 11973269, 21305474).

Overall, based on our results and the reported role of SOX17 in the literature it appears reasonable to conclude that SOX17 is necessary for proper endoderm differentiation. Following the reviewer’s suggestion, we now state this more clearly in the main text (page 7, lines 202-207).

“Notably, homozygous knockout cells fail to induce GATA4 expression and show no activation of T-REX17 (Figure supplement 3J).”

6. The proposed model, that LNCSOX17 is not acting in cis and therefore potentially in trans, is well supported by several experiments. But one further strong piece of evidence for a trans-acting function would be if the LNCSOX17 mutant phenotype could be rescued by expressing LNCSOX17 ectopically. Have the authors tried this? With the appreciation that the long transcript and variable splicing might make this difficult.

The reviewer is correct that our data imply that *T-REX17* is a potential *trans*-acting molecule, but we did not make a major point out of that as our results mostly argue against a *cis* role and don’t provide direct evidence for a *trans* mechanism.

We tried to strengthen this point further, and cloned the full-length *T-REX17* (~22kb) into a PiggyBac vector and delivered it (~31kb) into *T-REX17* depleted cells. As a control, we also created an isogenic line transfected with an empty vector only (PB only) (see panel A in Author response image 4). We generated stably integrated ectopic *T-REX17* expressing hiPSCs. After targeting, we derived four independent isogenic iPSC clones, one for PB only integrations and three clones for PB *T-REX17* integrations (PB *T-REX17* #1, #2, #3). We then verified the expression levels of *T-REX17* in these clones and saw *T-REX17* is already expressed in hiPSCs at levels comparable to the WT endoderm (see panel C in the reviewer figure below). As indicated by CXCR4-FACS at day 5 of EN-differentiation, we unfortunately could not observe rescue of CXCR4 protein levels (see panel B in Author response image 4).

There are of course many reasons why this complex rescue experiment did not work. For instance, the ideal experimental setup would have included the transient expression of *T-REX17* starting at day 3 of EN differentiation (Figure 2A). However, transfection of adherent human EN-differentiating cells with a ~31kb construct is extremely challenging various chemically based approaches have been tried without success including Lipo Stem (Thermo), Lipo3000 (Thermo), LipoLTX (Thermo), FuGENE (Promega). Other factors beyond temporal control may include expression levels, co-transcriptional regulation and localization. As a result, we cannot use this experiment to further comment on the *trans-*function of *T-REX17* and only provide these attempts as Author response image 4.

**Author response image 4. sa2fig4:** PiggyBAC (PB) integrated constitutive T-REX17 rescue. (A) Schematic of the ectopic PB *T-REX17* construct. Lines indicate randomly integrated sequence context for empty backbone (PB only) or *T-REX17* (PB *T-REX17*). (B) FACS histograms showing percentages of CXCR4^+^ cells during EN differentiation of sgT-REX17 dCas9-KRAB-MeCP2 cells of PB only or PB *T-REX17* clones. Clone numbers are indicated. Sample sizes are normalized to 9000 cells /sample. Two independent experiments are displayed. (C) qRT-PCR showing NANOG, SOX17 and *T-REX17* expression in PSCs and EN cells of sgT-REX17 dCas9-KRAB-MeCP2 cells integrated for either with PB only or PB *T-REX17*. Clone numbers are indicated. Fold change is calculated relative to the 18s housekeeping gene. Bar indicate the means, two independent experiments are represented. *T-REX17* wild type levels are indicated by the dashed line. Note, ectopic *T-REX17* is constantly expressed throughout differentiation.

7. I could not really follow why the transcript was called LNCSOX17. Presumably because it lies within the same TAD as SOX17? Nevertheless, I feel this name could be confusing because, as the authors show, LNCSOX17 does not have much to do with SOX17. The authors might want to consider proposing an alternative name for their transcript to avoid confusion over inferring a genetic or functional link to SOX17.

The name *LNCSOX17* derive from the fact that the lncRNA overlaps the SOX17 enhancer (*eSOX17*), it follows a similar gene expression pattern during endoderm formation, it is localized within the same loop domain (as the only other gene) and its expression is dependent on SOX17 function. Despite this series of crucial observation, we agree with the reviewer and the editors that, given that most lncRNAs overlapping with developmental enhancers regulate their cognate genes in cis-, the current name might be misleading.

Therefore, we changed the transcript name to *T-REX17* (Transcript Regulating Endoderm and activated by soX17) to still include some information about where it is transcribed, how it is regulated and its necessity for endoderm formation. We adapted the text, figures, tables accordingly.